# Highly Multiplexed Serology for Nonhuman Mammals

Alexa Schuettenberg,[a] Alejandra Piña,[a] Morgan Metrailer,[a] Ronald Guillermo Peláez-Sánchez,[b] Piedad Agudelo-Flórez,[b] Juan Álvaro Lopez,[c] Luke Ryle,[a] Fernando P. Monroy,[d] John A. Altin,[e] Jason T. Ladner[a,d]

aThe Pathogen and Microbiome Institute, Northern Arizona University, Flagstaff, Arizona, USA
bBasic Science Research Group, Graduate School, CES University, Medellín, Colombia
cMicrobiology School, Primary Immunodeficiencies Group, University of Antioquia, Medellín, Colombia
dDepartment of Biological Sciences, Northern Arizona University, Flagstaff, Arizona, USA
eThe Translational Genomics Research Institute (TGen), Flagstaff, Arizona, USA

**ABSTRACT** Emerging infectious diseases represent a serious and ongoing threat to humans. Most emerging viruses are maintained in stable relationships with other species of animals, and their emergence within the human population results from cross-species transmission. Therefore, if we want to be prepared for the next emerging virus, we need to broadly characterize the diversity and ecology of viruses currently infecting other animals (i.e., the animal virosphere). High-throughput metagenomic sequencing has accelerated the pace of virus discovery. However, molecular assays can detect only active infections and only if virus is present within the sampled fluid or tissue at the time of collection. In contrast, serological assays measure long-lived antibody responses to infections, which can be detected within the blood, regardless of the infected tissues. Therefore, serological assays can provide a complementary approach for understanding the circulation of viruses, and while serological assays have historically been limited in scope, recent advancements allow thousands to hundreds of thousands of antigens to be assessed simultaneously using <1 $\mu$L of blood (i.e., highly multiplexed serology). The application of highly multiplexed serology for the characterization of the animal virosphere is dependent on the availability of reagents that can be used to capture or label antibodies of interest. Here, we evaluate the utility of commercial immunoglobulin-binding proteins (protein A and protein G) to enable highly multiplexed serology in 25 species of nonhuman mammals, and we describe a competitive fluorescence-linked immunosorbent assay (FLISA) that can be used as an initial screen for choosing the most appropriate capture protein for a given host species.

**IMPORTANCE** Antibodies are generated in response to infections with viruses and other pathogens, and they help protect against future exposures. Mature antibodies are long lived, are highly specific, and can bind to their protein targets with high affinity. Thus, antibodies can also provide information about an individual's history of viral exposures, which has important applications for understanding the epidemiology and etiology of disease. In recent years, there have been large advances in the available methods for broadly characterizing antibody-binding profiles, but thus far, these have been utilized primarily with human samples only. Here, we demonstrate that commercial antibody-binding reagents can facilitate modern antibody assays for a wide variety of mammalian species, and we describe an inexpensive and fast approach for choosing the best reagent for each animal species. By studying antibody-binding profiles in captive and wild animals, we can better understand the distribution and prevalence of viruses that could spill over into humans.

**KEYWORDS** antibodies, emerging viruses, FLISA, highly multiplexed serology, PepSeq, serology, aichivirus, coronavirus, erbovirus, mammals, pestiviruses, virosphere

Address correspondence to Jason T. Ladner, jason.ladner@nau.edu.

The authors declare no conflict of interest.

Emerging and reemerging infectious diseases (especially those caused by viruses) represent a serious and ongoing threat to the human population. The last couple decades have offered many striking examples of this (e.g., Ebola virus disease [1], Hendra virus disease [2], Middle East respiratory syndrome [3], coronavirus disease 2019 [4], and monkeypox [5]), and both the frequency and the impact of emerging infectious diseases are likely to increase in the future due to a variety of factors, including globalization, increases in human population density, and the various impacts of climate change (6–8). However, we do not know which viruses pose the greatest future risk, which complicates efforts aimed at prevention and mitigation.

One thing that we do know is that the vast majority of emerging viruses are maintained in stable relationships with other species of animals, and emergence within the human population results from cross-species transmission facilitated by close contact between humans and animals (9). Therefore, if we want to be prepared for the next emerging virus, we need to broadly characterize the diversity and ecology of the viruses currently infecting other animals (i.e., the animal virosphere) (10). High-throughput metagenomic sequencing has accelerated the pace of virus discovery by enabling the deep and broad characterization of nucleic acids (11). However, molecular assays can detect only active (or latent) viral infections and only if virus is present within the sampled fluid or tissue at the time of collection. Therefore, in the context of understanding the animal virosphere, molecular surveillance is akin to searching for a needle in a haystack.

In contrast, serological assays measure long-lived antibody responses to infections, which can be detected within the blood, regardless of the infected tissues. Therefore, serological assays can provide a complementary approach for understanding the circulation of viruses within captive- and wild-animal populations, and while serological assays have historically been limited in scope (i.e., measuring antibody reactivity against a single antigen at a time), recent advancements now allow thousands to hundreds of thousands of antigens to be assessed simultaneously using $<1$ $\mu$L of blood (12, 13). These approaches for highly multiplexed serology have enabled the virome-wide characterization of exposure histories in humans (12), and they also offer the potential for deconvoluting cross-reactive antibody responses (13). However, to our knowledge, highly multiplexed serology has not been used to broadly assess antiviral antibody reactivity in nonhuman animals.

Several approaches for highly multiplexed serology have been described (e.g., PepSeq [13, 14], phage immunoprecipitation sequencing [PhIP-Seq] [15], and peptide arrays [16]), but they all depend on the availability of reagents that can be used to capture or label antibodies of interest. For example, for both PepSeq and PhIP-Seq, magnetic-bead-bound proteins are used to capture antibodies of interest. This step is critical as it enriches antibody-bound antigens, and it is this enrichment that is used to quantify binding. Similarly, peptide arrays also require proteins that can bind to conserved regions on the antibodies of interest; in this case, these proteins are used to fluorescently label antibodies that have bound to antigens printed on a solid surface. Therefore, the primary limitation of the application of highly multiplexed serology to nonhuman animals is the availability of appropriate antibody-binding proteins.

The most common antibody (or immunoglobulin [Ig])-binding proteins used for both PepSeq and PhIP-Seq bind to human IgG isotype antibodies using IgG-binding domains that were derived from bacterial proteins. These proteins play a critical role in immune evasion during bacterial infections, and they have been shown to bind to IgG from a variety of mammalian species, including humans (17, 18). Because of this characteristic, several of these bacterial IgG-binding domains have been adapted for use in molecular biology. These include domains from staphylococcal protein A (pA) and streptococcal protein G (pG), both of which are available commercially, on their own and in combination (pAG).

In order to evaluate the potential for these commercial immunoglobulin-binding proteins to enable highly multiplexed serology in various species of mammals, we first developed a competitive fluorescence-linked immunosorbent assay (FLISA) that is able

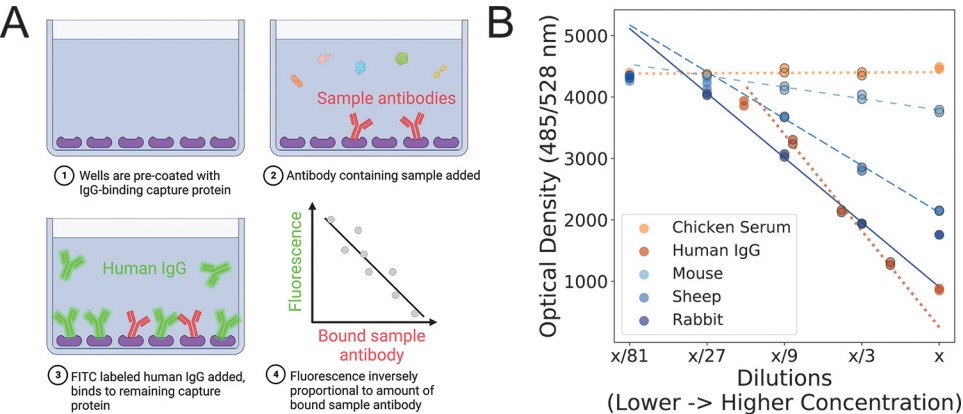

**FIG 1** Competitive FLISA provides a semiquantitative measure of the binding affinity between immunoglobulin-binding proteins and sample antibodies. (A) Diagram depicting the basic steps of the competitive FLISA presented here. Purple shapes represent immunoglobulin capture proteins. Unlabeled sample antibodies are shown in red, while fluorescently labeled control antibodies (known to have a strong affinity for the capture protein) are shown in green. The differently shaped and colored molecules in step 2 represent various other, nontarget proteins contained within the complex sample used as the input for the assay. (B) Example of the results from a single protein A (pA) FLISA plate, with negative (chicken serum) and positive (human IgG) controls shown in shades of orange and experimental samples (mouse ascitic fluid and sheep and rabbit sera) shown in shades of blue. Every sample was run at five dilutions, with two replicates per dilution, and "x" represents the highest concentration, which varied among the samples. The data points chosen for calculating the slope are outlined in gray, and the best-fitting line through those data points is plotted. For control samples, the best-fit slopes are shown as dotted lines. For the experimental samples, the degree of spacing between line segments is related to the estimated slope. A steeper slope indicates a stronger affinity between the sample antibodies and pA. Created with BioRender.com.

to semiquantitatively estimate binding affinity directly from complex, antibody-containing samples like serum and plasma. We then used PepSeq to demonstrate the utility of these commercial proteins for running highly multiplexed serology assays on a wide variety of mammalian species and the utility of the FLISA as an initial screen by comparing the relative FLISA binding affinities to PepSeq-based quantitative measures of antigen enrichment.

## RESULTS

**FLISA-based estimates of binding affinity.** In total, we used our FLISA (Fig. 1) to evaluate the efficacy of pA, pG, and pAG capture proteins across 66 samples from 26 mammalian species (including humans). Each sample was run against 1 to 3 capture proteins (on average, 2.9 capture proteins tested per sample), and at least one sample per species was run against all 3 capture proteins (see Table S1 in the supplemental material). Positive slope ratios (PSRs), a measure of binding affinity, ranged from 0.019 to 0.962, 0.004 to 1.492, and 0.064 to 0.908 for pA, pG, and pAG, respectively, and PSR values were generally consistent across different samples from the same species (Fig. 2). When considering PSRs from all species for which multiple samples were run against the same capture protein (56 data points across 18 species for each capture protein), we observed PSR standard deviations of 0.047, 0.109, and 0.064 for pA, pG, and pAG, respectively.

To assess our ability to measure relative binding affinities directly from complex sample types (e.g., serum, plasma, and ascitic fluid), we compared our FLISA PSR values to previously reported measures of relative affinity using purified IgG (11 mammalian species) (Fig. 3A; Table S1). For all three capture proteins, we observed significant correlations between our species-level average PSR values and IgG-specific inhibition data reported previously by Eliasson et al. (17) (Pearson correlation $P$ values of 0.004, 0.003, and 0.022 for pA, pG, and pAG, respectively). As expected, the observed correlations were in the negative direction (Fig. 3A), reflecting the fact that the PSR is positively correlated with the relative binding affinity, while the amount of target species IgG needed

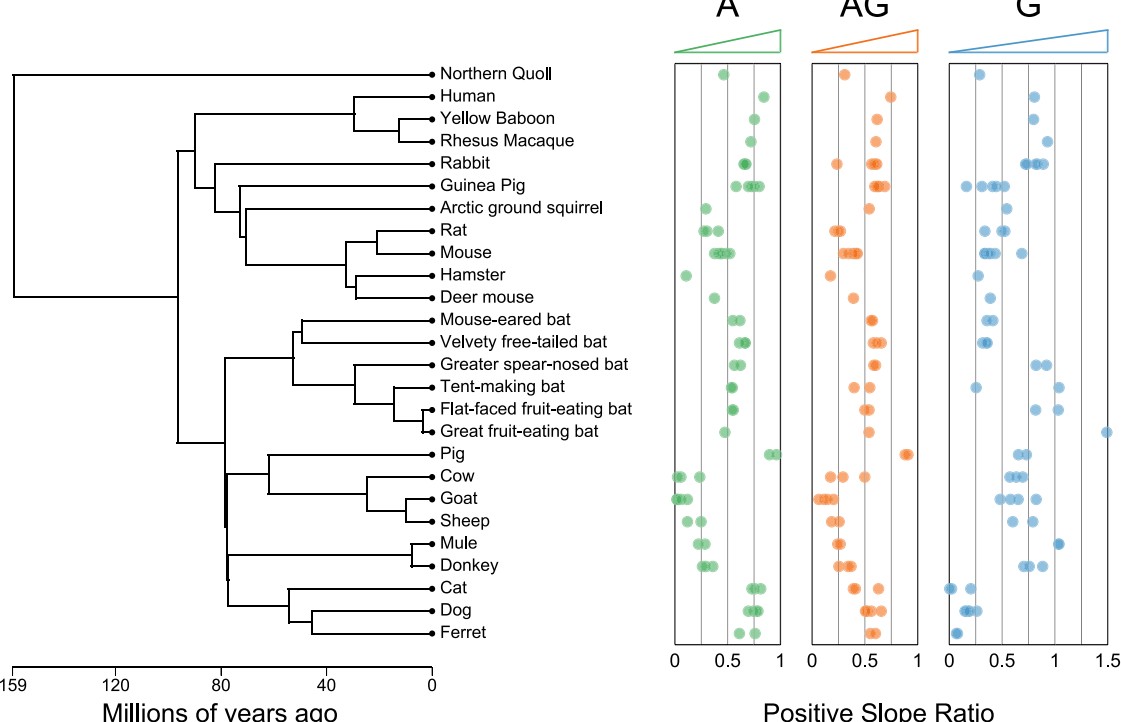

**FIG 2** Evolutionarily related species exhibit similar binding profiles. FLISA-based positive slope ratios for protein A (green), protein AG (orange), and protein G (blue) are shown. Each point represents a unique sample-capture protein combination, with samples from the same species shown at the same vertical position. Between 1 and 6 samples were assayed per species-capture protein combination. Species are oriented along the *y* axis according to the phylogenetic tree shown on the left, which was generated using TimeTree (55) on 23 July 2021. Three substitutions were made when generating the phylogenetic tree because of missing species in the TimeTree database (*Uroderma bilobatum* in place of *Uroderma convexum* ["tent-making bat"], *Myotis nigricans* in place of *Myotis* cf. *caucensis* ["mouse-eared bat"], and *Equus ferus* in place of mule). See Fig. S1 in the supplemental material for the correlation between evolutionary divergence and the pairwise difference in the positive slope ratio. For the scientific names of each species, see Table S1.

for 50% inhibition of rabbit IgG (17) is negatively correlated with the relative binding affinity.

All of the species tested exhibited average PSR values of >0.27 for at least one of the capture proteins, and ~81% (21/26) exhibited at least one average PSR value of >0.54. The lowest maximum PSR values, across the three capture proteins, were for the four tested species of muroid rodents (hamster, deer mouse, mouse, and rat) (0.273 to 0.454), followed by the one tested marsupial (Northern quoll) (0.463). There was no correlation between species-level PSR values for pA and pG (Pearson correlation *P* value of 0.71), but PSR values exhibited some correlation with phylogeny (Fig. 2; Fig. S1). Specifically, species that were closely related phylogenetically tended to exhibit similar PSR values. For example, all members of the Carnivora order (cat, dog, and ferret) exhibited high PSRs for pA (0.61 to 0.81 [*n* = 9]) but low PSRs for pG (0.004 to 0.26 [*n* = 10]). In contrast, all members of the Bovidae (cow, goat, and sheep) and Equidae (donkey and mule) families exhibited low PSRs for pA (0.02 to 0.36 [*n* = 15]) but high PSRs for pG (0.48 to 1.04 [*n* = 14]). All of the bat species tested exhibited moderately high PSRs for both pA (0.47 to 0.67 [*n* = 12]) and pG; however, the three species of the Phyllostomidae family exhibited markedly higher pG PSRs (0.82 to 1.5, with one outlier at 0.25 [*n* = 7]) than those of species of the Molossidae and Vespertilionidae families (0.32 to 0.41 [*n* = 5]).

The average PSR values for pAG were generally intermediate between the PSRs for pA and pG (Fig. S2). For 68% of the tested nonhuman mammal species (17/25), the average PSR for pAG was intermediate between the average PSRs for pA and pG. For

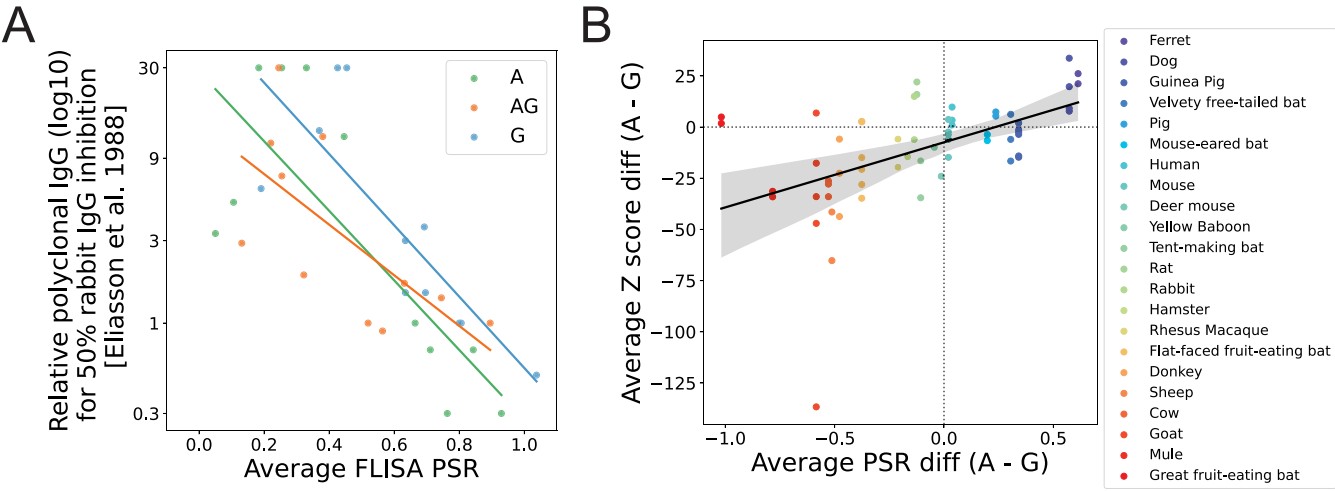

**FIG 3** The FLISA-based positive slope ratio (PSR) from complex samples correlates with the relative binding affinity of purified IgG and peptide-level enrichment in highly multiplexed serology assays. (A) Scatterplot comparing our FLISA-based PSR estimates from unfractionated plasma/serum/ascitic fluid (x axis) with previously reported (17) relative affinity estimates from purified polyclonal IgG. Each point represents a unique species-capture protein combination, with 11 species represented per capture protein (cow, dog, goat, guinea pig, horse/mule, human, mouse, pig, rabbit, rat, and sheep). Our FLISA data from mules were compared to the horse results reported previously by Eliasson et al. (17). PSR values represent the averages across samples when multiple samples from the same species were analyzed (see Table S1 in the supplemental material). Lines represent best-fit linear regressions generated using the regplot function in the seaborn python module (Pearson correlation P values of 0.004, 0.02, and 0.002 for proteins A, AG, and G, respectively). (B) Scatterplot comparing the difference (protein A − protein G) in the average PSR (x axis) to the average Z score difference (protein A − protein G) for peptides enriched above a threshold within a PepSeq assay using either capture protein. Each point represents a single sample assayed by both FLISA and PepSeq using both protein A and protein G as capture proteins. The solid line represents the best-fit linear regression generated using the regplot function in the seaborn python module, with the 95% confidence interval (1,000 bootstraps) shown by the gray ribbon (Pearson correlation P value of $1.38 \times 10^{-5}$). Dotted lines are drawn at values of zero on the x and y axes for reference.

another 28% (7/25), the average PSR for pAG was lower than the average PSRs for both pA and pG.

**Highly multiplexed serology using PepSeq.** In order to evaluate the potential for pA and pG to enable highly multiplexed serology for different species of mammals and to determine whether the PSR is a good predictor of capture protein performance, we assayed (i) 65 samples (from 24 species, including humans) with both pA and pG using at least one of our PepSeq libraries (HV1 [human virome version 1] and PCV [pancorona-virus]) and (ii) an additional 11 samples (from 8 species) using a single capture protein (pA or pG) (see Table S1 in the supplemental material for details). In total, we conducted at least one PepSeq assay for all of the nonhuman mammal species tested with our FLISA (n = 25), and we detected ≥1 enriched peptide for at least one sample-library combination for every species (interactive scatterplots highlighting the enriched peptides for each sample have been deposited in OSF [https://osf.io/8hygc/]).

To determine whether the FLISA PSR is a good predictor of capture protein performance, we compared PSR values to PepSeq Z scores, which provide a quantitative measure of enrichment for each peptide antigen. For this comparison, we included 61 sample-library combinations that were assayed with both pA and pG (including samples from 22 species) and for which we observed ≥5 enriched peptides (see Table S1 for details). Overall, we observed a significant positive correlation between the relative capture protein binding affinities (pA − pG) as measured using our FLISA (average species-level PSR difference) and PepSeq (average sample-level Z score difference) assays (Pearson correlation coefficient = 0.525; P = 1.38e−5) (Fig. 3B). There were, however, some notable outliers to this overall trend. For the rat samples (n = 2), we estimated a somewhat higher binding affinity for pG with our FLISA but greater PepSeq enrichment with pA. Similarly, we estimated a much higher pG affinity for the great fruit-eating bat (*Artibeus lituratus*) but saw slightly higher PepSeq enrichment with pA (n = 1 and 2 for FLISA and PepSeq, respectively).

We also assayed several samples using combinations of the IgG-binding domains from pA and pG. Specifically, we tested two approaches: (i) a commercially available

recombinant protein (pAG) that consists of fused IgG-binding domains from both pA and pG (Fig. S3A) and (ii) a mixture of individual pA- and pG-bearing magnetic beads (Fig. S3B). In some cases, the enrichment signals were comparable between assays that used a combination of the IgG-binding domains and those that used just pA or pG (e.g., pA versus pAG-agarose for sample NR-19260) (Fig. S3A). However, in other cases, the enrichment signal was substantially higher using either pA or pG in isolation (e.g., pG versus pAG for NR-3120) (Fig. S3A).

Of the 76 samples assayed using PepSeq, 29 came from nonhuman mammals that had been experimentally immunized with a viral antigen that is covered by peptides in one or both of our PepSeq libraries. These included immunogens of various complexities (e.g., full viral particles and individual proteins) derived from 15 different virus species (see Table S1 for details). In ~86% of these samples (25/29), we detected at least one enriched peptide from the known immunogen, with an average of ~28 enriched peptides per immunogen when considering only the library-capture protein combination with the greatest number of enriched peptides for each sample. To finely map epitopes and examine the cross-reactivity of antibodies elicited by immunization, we aligned homologous protein sequences from several viral congeners contained within our HV1 library and examined the distribution of reactive peptides along the alignments (Fig. 4A and B). NR-9404 is goat serum collected following immunization with the E1-E2 glycoprotein of Venezuelan equine encephalitis virus (VEEV). Our assays with HV1 identified 44 enriched VEEV peptides, which included both previously characterized and novel antibody epitopes, as well antibody reactivities that were specific for VEEV and those that appeared to exhibit cross-reactivity with peptides from 1 to 8 other virus species of the *Alphavirus* genus (Fig. 4A). Similarly, NR-9676 is deer mouse serum collected following immunization with the nucleocapsid protein of Sin Nombre virus (SNV). For NR-9676, we detected antibody reactivity against 9 SNV peptides from the nucleocapsid, which together represented ≥6 distinct epitopes, only 1 of which was specific for SNV (i.e., antibodies recognized only SNV peptides at that epitope) (Fig. 4B). Additionally, five of the HV1 assay samples were from goats and sheep that had been experimentally immunized with hemagglutinin proteins from different subtypes of influenza A virus (H1, H2, H5, H9, and H12). For each of these samples, we used our PepSeq data to calculate a relative enrichment score for hemagglutinin subtypes 1 to 14, and in each case, the highest relative enrichment score corresponded to the subtype used for immunization (Fig. 4C).

In our PepSeq assays, we also detected strong signals of antibody reactivity (≥5 enriched peptides) against many other viruses beyond those that were documented as known immunogens, and most of these reactivities are consistent with known host-virus associations. For example, we observed antibodies against erbovirus A peptides in samples from both a donkey and a mule (32 and 13 enriched peptides, respectively). Our analysis identified epitopes within the P1, P2, and P3 regions of the proteome, most of which were not previously reported in the Immune Epitope Database (IEDB), and many of these epitopes were shared between the two samples (Fig. 5A). We also observed antibody reactivity against betacoronavirus 1 peptides in samples from two Bovidae species (cow and goat) and two Equidae species (donkey and mule) with no documented history of infection/immunization (6 to 12 enriched peptides per sample). Strikingly, we observed strong congruence between the antibody epitopes observed in these samples and those observed in samples from animals with a documented history of immunization (Fig. 5B). Similarly, we observed antibody reactivity against (i) aichivirus A in two dogs (7 to 8 enriched peptides), three goats (15 to 21 enriched peptides), and a hamster (10 enriched peptides); (ii) mammalian orthoreovirus in three goats (7 to 10 enriched peptides) and one bat (5 enriched peptides); (iii) influenza A virus in a donkey and a mule (5 to 8 enriched peptides); and (iv) all eight species of the *Enterovirus* genus (enteroviruses A to E and rhinoviruses A to C) in three goats (107 to 174 enriched peptides summed across the eight virus species).

Finally, even in the absence of a well-established link between host and virus, we were able to generate strong evidence for potential past exposures based on (i) the

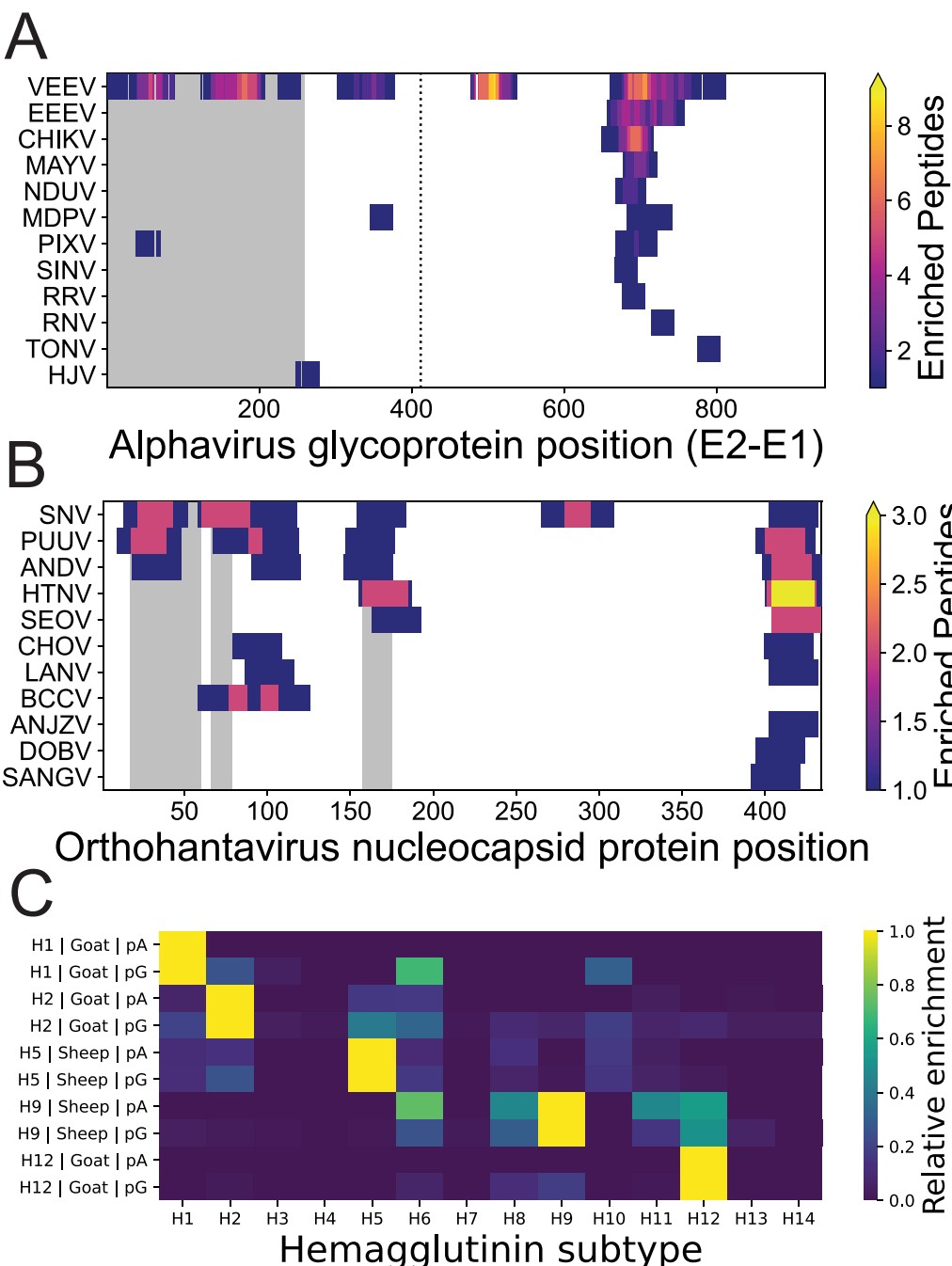

**FIG 4** Epitope-resolved antibody-binding profiles against experimental immunogens. (A and B) Distribution of enriched peptides (contained within the HV1 PepSeq library) across alignments of E2-E1 glycoproteins of several species in the *Alphavirus* genus from PepSeq assays (pA and pG) of goat serum (NR-9404) collected following experimental immunization with the Venezuelan equine encephalitis virus (VEEV) TC-83 (subtype IA/B) glycoprotein (A) and nucleocapsid proteins of several species in the *Orthohantavirus* genus from PepSeq assays (pA and pG) of deer mouse serum (NR-9676) collected following experimental immunization with the nucleocapsid protein of the SN77734 strain of Sin Nombre virus (SNV) (B). In panels A and B, each row represents assay peptides designed from a different virus species, and gray rectangles represent regions containing known epitopes from the species used for immunization (SNV) as documented in the Immune Epitope Database (IEDB). The vertical, black dotted line in (A) represents the break point between E2 and E1. (C) Relative enrichment scores for 14 subtypes of influenza A virus hemagglutinin (H1 to H14) calculated using PepSeq Z scores from assays of serum (NR-3148 [goat, H1], NR-4523 [goat, H2], NR-622 [sheep, H9], and NR-19222 [goat, H12]) or purified immunoglobulin (NR-49241 [sheep, H5]) from five animals following experimental immunization with a specific subtype of influenza A virus hemagglutinin. Each row represents a single sample-capture protein combination. The hemagglutinin subtype used for immunization, mammalian species, and capture protein are all indicated by the *y* axis labels. Additional abbreviations in panel A: CHIKV, chikungunya virus; EEEV, Eastern equine encephalitis virus; HJV, Highlands J virus; MAYV, Mayaro virus; MDPV, Mosso das Pedras virus; NDUV, Ndumu virus: PIXV, Pixuna virus; RNV, Rio

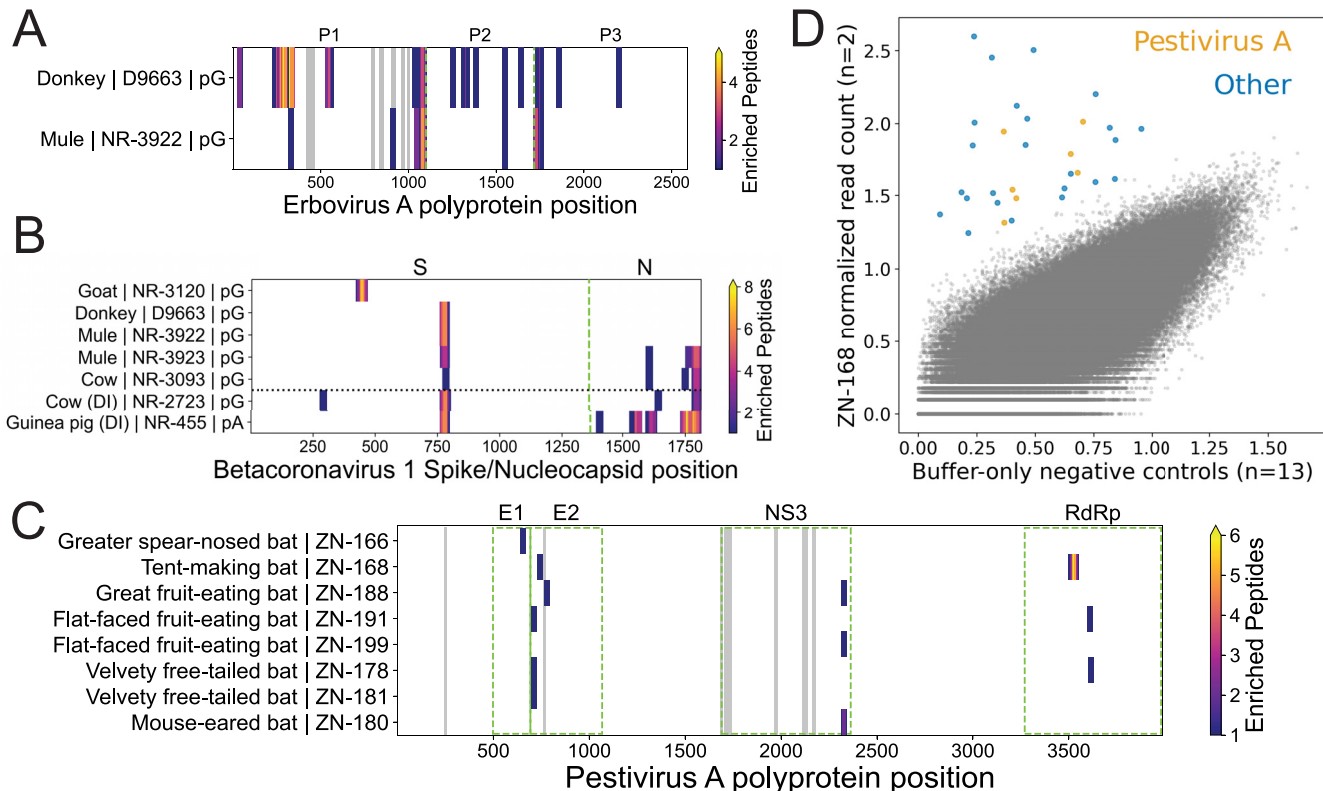

**FIG 5** Epitope-resolved antibody-binding profiles against viruses not associated with known infections/immunizations. (A to C) Distribution of enriched peptides (contained within the HV1 PepSeq library) across the erbovirus A polyprotein (A), betacoronavirus 1 spike and nucleocapsid proteins (B), and pestivirus A polyprotein (C). The y axis labels indicate the mammalian species of origin, the sample identifier, and, for panels A and B, the capture protein used for the PepSeq assay. For panel C, both pA and pG assays were considered for each sample, when available. In panels A and C, the gray rectangles represent regions containing known epitopes as documented in the Immune Epitope Database (IEDB). Green dashed lines delineate labeled protein regions, while the black dotted line in panel B separates hosts with documented betacoronavirus 1 immunizations ("DI") (bottom) from those without documented immunizations (top). RdRp, RNA-dependent RNA polymerase. (D) Scatterplot comparing normalized read counts from HV1 library PepSeq assays of buffer-only negative controls (x axis) (13 replicates) and a serum sample from a tent-making bat (ZN-168) (y axis) (2 replicates). Each circle represents a unique peptide. Enriched peptides are shown as larger, colored circles, while nonenriched peptides are shown in smaller, gray circles. The seven orange circles represent enriched peptides from pestivirus A (see panel C for locations within the polypeptide), while all other enriched peptides are shown in blue.

enrichment of multiple peptides overlapping the same epitope; (ii) the enrichment of multiple epitopes from the same virus species, genus, or family; and/or (iii) consistent patterns of enrichment across multiple individuals from the same host group. For example, we observed antibody reactivity against peptides designed from pestivirus A in several of the bat sera that we tested (Fig. 5C and D). In total, we observed enriched pestivirus A peptides in our assays of sera from eight different bats, representing six different species (1 to 7 enriched peptides per individual). These peptides clustered within four different proteins and, in total, represented 7 epitope regions (1 to 2 epitopes per individual), 3 of which were recurrent across multiple individuals (Fig. 5C).

Notably, three of the assayed samples consistently (across replicates, capture proteins, and PepSeq libraries) resulted in high numbers of enriched peptides compared to the other samples in this study. These samples included one plasma sample from a baboon (SNPRC-025 [pA capture protein/HV1 PepSeq library = 821 enriched peptides; pG/HV1 = 939]), one serum sample from a ferret (NR-19264 [pA/HV1 = 1,308; pA/PCV = 421]), and one ascitic fluid sample from a mouse (NR-48961 [pA/HV1 = 2,965; pG/HV1 = 1,571; pG/PCV = 480]). These enriched peptide counts are 31 to 152 times higher than the median values that we observed for nonhuman samples with the cor-

**FIG 4** Legend (Continued)

Negro virus; RRV, Ross River virus; SINV, Sindbis virus; TONV, Tonate virus. Additional abbreviations in panel B: ANDV, Andes orthohantavirus; ANJZV, Anjozorobe orthohantavirus; BCCV, Black Creek Canal orthohantavirus; CHOV, Choclo orthohantavirus; DOBV, Dobrava-Belgrade orthohantavirus; HTNV, Hantaan orthohantavirus; LANV, Laguna Negra orthohantavirus; PUUV, Puumala orthohantavirus; SANGV, Sangassou orthohantavirus; SEOV, Seoul orthohantavirus.

responding PepSeq libraries and 1.9 to 7.8 times higher than the next-highest enriched peptide count for a nonhuman sample (excluding the counts for these three samples). Given that the virus exposure histories for these samples are incompletely documented, we cannot be certain whether all of the enriched peptides are reflective of true exposure to the viruses targeted in our assays or whether there could another reason for the high level of reactivity, such as the presence of highly cross-reactive antibodies. Therefore, out of an abundance of caution, we have not included these three samples in our virus-level descriptions of antibody reactivity.

## DISCUSSION

Approaches for highly multiplexed serology (e.g., PepSeq, PhIP-Seq, and peptide arrays) enable the simultaneous assessment of antibody binding to thousands to hundreds of thousands of peptide antigens from $<1$ $\mu$L of blood (12, 13), and because antibodies can serve as long-lived biomarkers of past infections, we can use highly multiplexed serology to better understand the diversity and ecology of the viruses that infect nonhuman animals, and which, therefore, may have the potential to spill over into the human population. Here, we evaluate the utility of several commercial IgG-binding proteins for enabling highly multiplexed serology in a wide variety of mammalian species, and we describe a simple, competitive FLISA that can be used as an initial screen to help identify the best capture protein to use for any mammalian species of interest.

Recombinant versions of two bacterial proteins, staphylococcal protein A (pA) and streptococcal protein G (pG), have been developed into commercial reagents and are widely used for immunological applications, including the immunoprecipitation of antibodies to facilitate liquid-phase approaches to highly multiplexed serology (e.g., PepSeq and PhIP-Seq) (13, 15). The immunoglobulin-binding domains of pA and pG have both been shown to bind to IgGs from a variety of mammalian species but often with very different affinities (17, 18), and to our knowledge, our study represents the first demonstration that this binding is sufficient to enable highly multiplexed serology in nonhuman animals.

Using the PepSeq platform (13, 14), we demonstrate that both pA and pG have the potential to broadly enable highly multiplexed serology in mammalian hosts but also that assay sensitivity is related to the relative affinity with which these proteins bind antibodies from different species. One of our goals was to design a simple assay that could accurately assess the relative affinity of capture proteins for antibodies from a variety of animal species, and we wanted the assay to work directly with serum/plasma (i.e., not requiring purified antibodies) and to be robust to the initial antibody concentration within the sample. To accomplish these goals, we utilized a multiple-dilution competitive FLISA approach (Fig. 1), with the capture protein of interest fixed to the bottom of each well and fluorescein isothiocyanate (FITC)-labeled human IgG used as a reporter. We chose human IgG as our reporter because pA and pG are both known to exhibit strong binding affinities for human IgG, and both proteins have been shown to work well for immunoprecipitation-based highly multiplexed serology assays with human samples (12, 13, 19). Using this FLISA, we characterized samples from 26 mammalian species and demonstrated strong correlations between our FLISA-based affinities and previously reported relative affinities measured from purified IgG (Fig. 3A) as well as quantitative measures of enrichment (Z scores) from our PepSeq assays (Fig. 3B). These results demonstrate the utility of our FLISA for (i) estimating the relative affinities of pA and pG for antibodies directly from complex samples (e.g., serum, plasma, and ascitic fluid) and (ii) gauging the potential for these capture proteins to enable sensitive, highly multiplexed serology assays. It should be noted that our assay is not able to directly determine which isotypes are binding to the capture proteins. However, based on known isotype-binding profiles for these proteins (20, 21) and our strong correlations with measurements from purified IgG, we expect that most of the bound antibodies, across the various mammalian species, will be IgG.

Using pA and/or pG, we successfully detected antibody binding to peptide antigens within samples from all 25 of the tested nonhuman mammalian species, and these samples included representatives of 25 genera, 17 families, and 8 orders. The presence of enriched peptides demonstrated both (i) the successful immunoprecipitation of antibodies using pA and/or pG and (ii) antibody binding to peptide antigens present in our libraries. In fact, we observed significant peptide enrichment within PepSeq assays even with species-capture protein combinations that exhibited relatively low affinities in our FLISAs. For example, we measured PSR values for our hamster serum at 0.11 and 0.27 for pA and pG, respectively, yet we observed 6 and 72 enriched peptides when assaying this sample against our HV1 PepSeq library with pA and pG, respectively. However, we generally observed little, if any, enrichment in cases where the PSR was <0.1 (cow and goat for pA; cat and ferret for pG). Overall, we expect that future PepSeq libraries, designed specifically to cover viruses known to infect the nonhuman hosts of interest, will result in more enriched peptides and, therefore, improved estimates of assay performance.

In general, we observed similar patterns of affinity between antibodies and bacterial IgG-binding proteins for phylogenetically closely related mammalian species (Fig. 2; see also Fig. S1 in the supplemental material). This is consistent with differences in binding affinity being driven by genetic differences between species in the constant regions of the genes that encode the Ig heavy and light chains. This pattern also suggests that the relative binding affinity for untested mammalian species can be reasonably approximated by comparison to available data from close relatives. Additionally, by comparing the relative affinities of binding to Ig heavy and light chain amino acid sequences from a variety of species, we may be able to better understand the binding motifs of these bacterial IgG-binding proteins and therefore predict relative affinities even for species for which data for close relatives are unavailable.

Another potential approach for dealing with differences in the relative affinities of pA and pG across mammalian species is to combine these two proteins within a single assay. This could be done with recombinant pAG, which combines the IgG-binding sites of both pA and pG within a single construct, or with a mixture of pA and pG. Our initial tests of both approaches showed some promise, with these combinations sometimes being able to match the enrichment signal observed with the highest-affinity capture protein in isolation (Fig. S3). However, we observed quite a bit of variability in the relative performances of these combination approaches with different samples/species. More work is needed to understand whether this relative performance is species specific and whether these approaches can be further optimized. One important difference is the types of beads on which pAG is commercially available. Our standard protocol is optimized for 2.8-$\mu$m Dynabeads (Invitrogen), but Dynabeads do not currently offer a pAG option. For pAG, we have tested 1-$\mu$m magnetite-coated polymeric beads (catalog number 88802; Pierce) and 10- to 40-$\mu$m agarose beads (catalog number 78609; Pierce). It is also possible that there is some interference between antibody binding at the pA and pG IgG-binding domains on the recombinant pAG constructs, which impacts overall binding in these assays. Additionally, in order to ensure that the overall binding capacity was not reduced for species with a poor affinity for one of the capture proteins, our pA-plus-pG assays included twice the volume of beads compared to that used for our typical assays with pA or pG in isolation. Therefore, even if these assays were able to consistently match the performance of the individual pA or pG assays, they would be more expensive to run.

By characterizing several blood samples collected following experimental immunizations, we were able to demonstrate the power of highly multiplexed serology for carefully dissecting the immune responses against specific antigens, including the presence of antibodies that may cross-recognize homologous antigens from related viruses (Fig. 4). By simultaneously assaying hundreds of thousands of peptides, highly multiplexed serology can provide epitope-resolved data across a wide variety of antigens, and therefore, this approach may be useful for clinical diagnostics, either directly or for the discovery of

antigens that can then be adapted for use with less highly multiplexed approaches. However, in this study, we did not attempt to calculate measures of sensitivity or specificity for the detection of specific virus exposures/immunizations. This is because our sample size for each specific immunogen was very small and because our knowledge of the infection histories of the characterized samples was incomplete.

In addition to detecting the expected reactivities against documented immunogens (Fig. 4), we also observed many additional reactivities that are consistent with documented host-virus relationships. These include several members of the *Picornaviridae* family. For example, erbovirus A (i.e., equine rhinitis B virus), which is a member of the *Erbovirus* genus, is known to be a common respiratory pathogen of horses (22). We observed broad antibody reactivity against erbovirus A peptides in serum samples from both a donkey and a mule (Fig. 5A). We also observed antibody reactivity against aichivirus A peptides in samples from two dogs, three goats, and one hamster. Aichivirus A is a species within the *Kobuvirus* genus, and it includes (i) a virus that has been isolated, on multiple occasions, from diarrheic dogs (canine kobuvirus) (23, 24) and (ii) viruses that have been identified in association with many different species of rodents (murine and rat kobuviruses) (25–27). Although we are not aware of any examples of aichivirus A isolation from goats, several closely related kobuviruses have been reported in goats, including aichivirus B (28, 29) and aichivirus C (29, 30), neither of which was targeted by our PepSeq assays. We also detected the enrichment of mammalian orthoreovirus (family *Reoviridae*) peptides in three goats and one bat (*Myotis* cf. *caucensis*; Vespertilionidae family). Mammalian orthoreovirus is known to infect a broad range of mammalian species, and this viral species has been identified in both Eurasia and North America in association with several species of bats of the Vespertilionidae family (31–33). Within each of our seropositive samples, we observed reactivity against 3 to 5 of the 10 genome segments (collectively, we observed antibody reactivity against 8/10 segments).

Betacoronavirus 1, which is a member of the *Coronaviridae* family, includes a common human pathogen (human coronavirus OC43) as well as viruses that commonly infect a variety of nonhuman animal species (e.g., bovine coronavirus and equine coronavirus). We detected robust antibody reactivity against betacoronavirus 1 ($\geq$6 peptides) in five animals with no documented exposures, and the epitopes recognized in these individuals were similar to those observed following experimental immunization (Fig. 5B). Reactive samples were collected from a cow, a goat, a donkey, and two mules. To our knowledge, cows and goats are known hosts for bovine coronavirus (34–36), while horses and donkeys are known hosts for equine coronavirus (37–39). However, our data suggest that the host ranges of these viruses may be broader than currently documented. The most commonly recognized betacoronavirus 1 epitope is located in the spike protein, overlapping the S1/S2 cleavage site (Fig. 5B). By comparing the levels of reactivity at this epitope to those for homologous peptides from several different viruses, we found that the antibody reactivity in the donkey and mules was strongest against bovine coronavirus peptides, while the reactivity in the cow was strongest against an equine coronavirus peptide (Fig. S4) (antibodies within the goat sample did not recognize this epitope).

Also of note, we observed antibody reactivity against pestivirus A peptides in eight serum samples collected from bats in Necoclí, Colombia, including representatives of six species, five genera, and three families (Fig. 5C and D). In total, this included reactivity against 14 unique peptides, 2 of which were recognized by antibodies in multiple individual bats, and collectively, these peptides clustered into 7 epitopes. Four of these epitopes are within the E1 and E2 envelope glycoproteins (collectively recognized in six samples), and one is within the NS3 protein (collectively recognized in three samples). All of these proteins are known to be immunogenic (40–43). The two remaining epitopes fell within the RNA-dependent RNA polymerase (collectively recognized in three samples), which is one of the most highly conserved proteins across pestivirus species (Fig. S5). Pestivirus A is one of 11 recognized species in the *Pestivirus* genus and the only one of these species included in our assay (44). We are not aware of any reports of pestivirus A infections of bats; however, two currently unclassified lineages

of bat pestiviruses have been documented via high-throughput sequencing from two species (and two families) of bats in China (*Rhinolophus affinis* and *Scotophilus kuhlii*) (45, 46). Our results suggest that the geographic and taxonomic distributions of bat pestiviruses are substantially broader than currently recognized. Additionally, many of the pestivirus epitopes recognized by bat antibodies in this study are poorly conserved in the bat pestiviruses from China (Fig. S5), which are distantly related to pestivirus A (44). Therefore, these reactivities may represent infections with distinct bat pestivirus lineages.

## MATERIALS AND METHODS

**Samples.** In total, our study included 82 samples from 25 species of nonhuman animals (including mule, a sterile hybrid) as well as 4 human samples as controls. Samples were obtained from a variety of sources, including BEI Resources, commercial companies, and both academic and nonprofit organizations (for details, see Table S1 in the supplemental material). For the collection of the Northern quoll sample (BQ), all research methodologies were approved by the University of Queensland animal ethics committee (SBS/541/134 12/ANINDILYAKWA/MYBRAINSC) and were conducted under a permit provided by the Northern Territory Parks and Wildlife Commission (permit number 47603) and with the permission of the Traditional Owners of Groote Eylandt, the Anindilyakwa peoples. For the collection of the Arctic ground squirrel sample (BAGS), all procedures were approved by the University of Alaska—Fairbanks Institutional Animal Care and Use Committee (IACUC) (approval numbers 340270 and 864841) and were performed according to the guidelines established by the American Society of Mammalogists (47). Several of the cat and dog samples (713629, 713635, 713677, 713661, 713807, and 713761) were collected as described previously by Yaglom et al. (48), with the approval of the Translational Genomics Research Institute (TGen) IACUC (approval number 20163). Additional dog samples (3226 to 3231) were collected under a protocol approved by the University of Georgia IACUC (approval number A2017 05-014-Y1-A0). Two of the nonhuman primate samples (SNPRC-025 and SNPRC-027) were collected from animals involved in studies approved by the Texas Biomedical Research Institute IACUC. Three of the human samples (VW-128, VW-139, and VW-229) were collected at ValleyWise Health in Phoenix, AZ, with the approval of the ValleyWise Health Institutional Review Board and the Institutional Review Board of Northern Arizona University (approval number 1545420). The mouse serum sample (MNAU-sterile) was collected under a protocol approved by the IACUC of Northern Arizona University (approval number 16-008). The bat sera (ZN-165 to ZN-205) were collected with the permission of the Autoridad de Licencias Ambientales (ANLA) of the Colombian Ministry of the Environment (approval number 0790, 18 July 2014) (49).

**FLISA.** To measure the affinity of each capture protein for antibodies from nonhuman mammal species, we designed a competitive FLISA (Fig. 1A), which utilized a known strong binder (human IgG) as a reporter. For this assay, we precoated plates with the capture protein of interest, which included recombinant pA, pG, and pAG (Pierce-Thermo Fisher). For pA and pG, we used both commercially prepared (Pierce-Thermo Fisher) and in-house-coated plates; all pAG-coated plates were prepared in-house. For the in-house preparations, we added 300 ng of the capture protein (diluted in 0.05 M carbonate-bicarbonate coating buffer [pH 9.6]) to black MaxiSorp-treated 96-well immunoplates (Thermo Fisher) and then incubated the plates at 4°C for 16 h. Following incubation, we washed the wells three times with phosphate-buffered saline (pH 7.4) containing 0.05% Tween 20 (PBST). We then blocked the plates by adding 150 $\mu$L of SuperBlock T20 (Thermo Fisher) to each well, followed by a 1-h incubation at 37°C.

We then added dilutions of antibody-containing samples to the precoated plates and incubated the plates for 2 h at room temperature, followed by three washes, each with 150 $\mu$L of PBST, to remove unbound antibodies. Specifically, we added 100 $\mu$L of the sample diluted in PBST and assayed each sample at five serial dilutions. For experimental samples (e.g., serum/plasma), we began each series with a 1:100 dilution of the sample, followed by additional 1:3 serial dilutions. We included chicken serum (catalog number NR-3132; BEI Resources) on each plate as a negative control; all of the capture proteins that we tested have been reported to have no binding affinity for chicken immunoglobulin (17). We also included purified, normal human IgG (catalog number 5503; ProSci) on each plate as a positive control; all three capture proteins tested have high affinities for human IgG (17, 50). For each positive-control dilution series, we initially diluted the human IgG to 16 $\mu$g/mL (pG) or 32 $\mu$g/mL (pA and pAG) and then generated four additional 1:2 serial dilutions. Finally, we added 100 $\mu$L of FITC-conjugated human IgG (9 ng/$\mu$L, diluted in PBST) to each well, incubated the plates for 1 h at room temperature, and read the fluorescence at 485/528 nm (excitation/emission) after shaking on medium for 5 min using a BioTek Synergy HT microplate reader.

Each sample was assayed in duplicate within a single plate, and the relative affinity between the immobilized capture protein and sample antibodies was quantified as a "positive slope ratio" (PSR) using a custom python script (https://github.com/jtladner/Scripts/blob/master/FLISA/analyzeCompFLISA_dilutionFactors.py). The PSR is a measure of the relative affinity of sample antibodies compared to normal human IgG and was calculated by comparing the rate of change in fluorescence with sample dilution between each sample and the positive-control dilution series run on the same plate. Specifically, the rate of change for each sample was calculated using the three consecutive dilutions that resulted in the highest rate of change (Fig. 1B). Our goal was to capture the portion of the curve in which there is a linear relationship between the change in the sample concentration and the change in the measured fluorescence. The slope that we measured should be roughly equivalent to the Hill coefficient of a four-parameter logistic curve. However, we found that the linear fit was more broadly applicable across the range of affinities that we were measuring while using a

consistent dilution series across samples (Fig. 1B). Samples that exhibited poor linear correlations and/or low maximum fluorescence (compared to the negative control) were rerun after adjusting the dilution series appropriately. If we observed a poor positive-control linear correlation or a significant negative-control linear correlation, the entire plate was rerun.

**PepSeq.** PepSeq technology enables the production of diverse, fully defined libraries of DNA-barcoded peptides, which can be used to simultaneously assess antibody binding to thousands to hundreds of thousands of antigens. All PepSeq assays were performed according to the protocols detailed previously by Ladner et al. (13) and Henson et al. (14). In brief, 0.1 pmol of the PepSeq library (5 $\mu$L) was added to a diluted sample (0.5 $\mu$L of the sample in 4.5 $\mu$L of SuperBlock), and the mixture was incubated overnight. The binding reaction mixture was applied to 10 $\mu$L of prewashed pA- or pG-bearing magnetic beads and incubated for ≥15 min to capture antibodies within the sample. The beads were then washed, bound PepSeq probes were eluted by incubation at 95°C for 5 min, and the DNA portions of the PepSeq probes were amplified and prepared for Illumina sequencing using barcoded DNA oligonucleotides (13). Following PCR cleanup, products were pooled across assays, quantified, and sequenced using an Illumina NextSeq instrument (75- or 150-cycle kit). Two slightly different washing procedures were employed for different assay plates, one using a handheld magnetic bead extractor and the other involving manual pipetting. However, for all direct comparisons between capture proteins for the same sample, we utilized a consistent plate-washing procedure. We also ran a limited number of PepSeq assays that combined IgG-binding domains of pA and pG within a single assay. We tested two approaches for this: (i) the use of a combination of pA- and pG-bearing magnetic beads (10 $\mu$L of each was added to each assay mixture, for 20 $\mu$L of beads in total) or (ii) the use of magnetic beads bearing recombinant pAG. For the pA-pG combination, PepSeq assays were run exactly as described above. We tested two different types of commercial beads containing recombinant pAG: 1-$\mu$m magnetic beads (catalog number 88802; Pierce) and 10- to 40-$\mu$m agarose magnetic beads (catalog number 78609; Pierce). Normalizing for differences in the binding capacity, we added 5 $\mu$L of 1-$\mu$m nonagarose beads and 2.5 $\mu$L of 10- to 40-$\mu$m agarose beads to each assay mixture. According to the manufacturer's recommendations, both types of pAG beads were incubated with each sample for 1 h prior to washing.

In this study, we utilized two different PepSeq assays: our 244,000-peptide human virome version 1 assay (HV1) and our 99,971-peptide pancoronavirus (PCV) assay. HV1 was designed to broadly cover potential linear epitopes from several hundred viruses known to infect humans and was previously described in detail by Ladner et al. (13). PCV was designed to broadly cover potential linear epitopes from all sequenced coronaviruses (including the alpha-, beta-, delta-, and gammacoronavirus genera) regardless of the host species (see File S1 for details). To ensure consistency, all sample-library-capture protein combinations were assayed in duplicate.

**PepSeq analysis.** PepSIRF v1.4.0 was used to analyze the sequencing data from the PepSeq assays (51). First, the data were demultiplexed, and reads were assigned to peptide antigens using the demux module of PepSIRF with up to one mismatch within each barcode (8 to 12 nucleotides [nt] in length) and up to three mismatches in each DNA tag (40 nt in length). A truncated, 40-nt DNA tag (full length = 90 nt) was used to enable sequencing with a 75-cycle kit, and we excluded 70 and 351 peptides from consideration for the HV1 and PCV libraries, respectively, as the first 40 nt of these DNA tags are not unique within these libraries.

Enrichment Z scores were then calculated using the diffEnrich pipeline as implemented in the auto-pepsirf QIIME 2 plug-in (https://ladnerlab.github.io/pepsirf-q2-plugin-docs/) (52, 53). This pipeline automates the analysis protocols described previously by Ladner et al. (13) and Henson et al. (14). Briefly, this protocol included normalization to control for differential sequencing depths across samples (norm PepSIRF module and col_sum method), normalization against buffer-only negative controls (norm PepSIRF module and diff method), and calculation of Z scores using a 95% highest-density interval with peptide bins that were based on 6 to 14 buffer-only negative controls (zscore PepSIRF module). There was no discernible difference between the buffer-only controls run with pA- and pG-bearing beads from assays with the same PepSeq library. Therefore, for each library, controls of both types were considered together for the formation of bins (bin PepSIRF module). Likewise, for the PepSeq comparisons of pA and pG with pAG, buffer-only controls run with all three capture proteins were used to generate the bins.

To ensure consistency in signals across replicates, Z score correlations were visualized using the repScatters module in the ps-plot QIIME 2 plug-in (https://ladnerlab.github.io/pepsirf-q2-plugin-docs/plugins/ps-plot/repScatters/) and manually inspected. For sample-library-capture protein combinations with poor Z score correlations, we either reran the relevant assays or omitted those assays from the presented data. To identify enriched peptides, we chose thresholds that minimized false-positive results when analyzing pairs of buffer-only negative controls that were not considered in the generation of bins or normalization. For PCV assays, we used a Z score threshold of 7.5, and for HV1 assays, we used a Z score threshold of 10, in combination with a column-sum-normalized read count (i.e., reads per million) threshold of 2. These thresholds resulted in averages of 0 and 0.19 enriched peptide calls in 6 and 21 pseudoreplicate pairs of buffer-only negative controls for PCV and HV1, respectively.

To compare assays conducted with the same sample and library but different IgG capture proteins, we calculated average Z scores for each assay across the union of enriched peptides from all assays being compared. Higher Z scores indicate greater enrichment within the assay. Therefore, we expect to see higher average Z scores for assays run with capture proteins that bind substantially better to antibodies from a given species. MAFFT v7.490 was used to align enriched peptides to full-length reference proteins (with the –keeplength and –addfragments options) and to align homologous reference proteins from different viral species (default conditions). For animals immunized with specific viral proteins (e.g., influenza A virus hemagglutinin), NCBI BLASTp v2.12.0$^+$ (54) was used to compare all assay peptides against a representative sequence of the appropriate immunogen, and any peptide with an E value of ≤1e−15 was

considered a match. In cases where a specific protein was not specified, we assumed that whole and/or infectious viral particles were used for immunization, and therefore, all peptides designed from the relevant virus species were considered to be matches.

**Data availability.** All relevant data are within the paper, are within the supplemental material, and/or have been deposited in OSF (https://osf.io/8hygc/).

## SUPPLEMENTAL MATERIAL

Supplemental material is available online only.
**SUPPLEMENTAL FILE 1**, PDF file, 1.6 MB.
**SUPPLEMENTAL FILE 2**, XLSX file, 0.03 MB.

## ACKNOWLEDGMENTS

We acknowledge the following people for their assistance in acquiring samples for this project: Loren Buck (NAU), Rachel Caballero (ValleyWise Health), Emily Cope (NAU), Luis Giavedoni (Trinity University), Sierra Jaramillo (NAU), Mary Mulrow (ValleyWise Health), Lora Nordstrom (ValleyWise Health), Dan Quan (ValleyWise Health), Chandler Roe (NAU), Heather Venkat (ADHS), Guilherme Verocai (UGA), Cory T. Williams (CSU), Robbie Wilson (University of Queensland), and Hayley Yaglom (TGen). For detailed acknowledgments of samples obtained from BEI Resources, see Table S1 in the supplemental material.

This work was supported by the State of Arizona Technology and Research Initiative Fund (TRIF) (administered by the Arizona Board of Regents through Northern Arizona University) and by the Ministerio de Ciencia Tecnología e Innovación of Colombia (122877757660 to P.A.-F.). The funders had no role in study design, data collection and analysis, decision to publish, or preparation of the manuscript.

We declare no competing interests.

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
