## [Reviewer comments · Microbiology Spectrum]

Microbiology Spectrum

Highly-multiplexed serology for non-human mammals

Alexa Schuettenberg, Alejandra Piña, Morgan Metrailler, Ronald Peláez-Sánchez, Piedad Agudelo-Flórez, Juan Lopez, Luke Ryle, Fernando Monroy, John Altin, and Jason Ladner

Corresponding Author(s): Jason Ladner, Northern Arizona University

Review Timeline:

Submission Date:	July 29, 2022
Editorial Decision:	August 23, 2022
Revision Received:	August 30, 2022
Accepted:	September 6, 2022

Editor: Ralph Tripp

Reviewer(s): Disclosure of reviewer identity is with reference to reviewer comments included in decision letter(s). The following individuals involved in review of your submission have agreed to reveal their identity: Swinburne A J Augustine (Reviewer #1)

Transaction Report:

DOI: <https://doi.org/10.1128/spectrum.02873-22>

August 23, 2022

Dr. Jason T Ladner
Northern Arizona University
1395 Knoles Drive
Coconino County
Flagstaff, AZ 86001

Re: Spectrum02873-22 (Highly-multiplexed serology for non-human mammals)

Dear Dr. Jason T Ladner:

Please respond in a point-by-point reply to the reviewer's comments.

Link Not Available

Sincerely,

Ralph Tripp

Journals Department
Reviewer comments:

Reviewer #1 (Comments for the Author):

Title: Highly-multiplexed serology for non-human mammals

Review

Abstract, Title, References:

The aim/objectives of the study should be made clear. The authors have "demonstrated" the potential for commercial immunoglobulin-binding proteins, protein A and protein G, to enable highly multiplexed serology in non-human animals using a competitive Fluorescence-Linked Immunosorbent Assay (FLISA). However, the aim of the study, although implicit in the Abstract and Introduction, should be clear. The authors clarified the objective "to evaluate the utility of..." later in the manuscript. That

should be mentioned much earlier. What is clear, is the study design, performance, and Results.

The title communicates the intent of the authors clearly and succinctly and is informative and relevant.

The references are relevant, recent, referenced correctly, and appropriate key studies are included. Since the monkeypox outbreak is new and ongoing, the authors could reference that outbreak to strengthen their argument for testing non-human mammals to identify current and emerging pathogens that could potentially impact public health.

Introduction:

The authors have clearly described the use and benefits of multiplexed serological testing in humans and identified a gap in applying these approaches to non-human subjects.

Given what is already known and the gaps in our knowledge, the research question is not only justified but essential to arresting future epidemics and pandemics, especially those associated with zoonotic infections by identifying antibody or therapeutic targets if these pathogens make the leap from animals to humans like the SARS-CoV-2 coronavirus.

Materials and Methods:

This section is clear. The authors have included enough detail in the M&M section that their experimental design can be replicated. All animal and human samples used in the study were IACUC- or IRB-approved by appropriate institutions.

Controls, sampling, statistical analyses, and reporting are well described and appropriate for the study.

Results:

Although the experimental design and statistical approaches are complex, the authors have presented the data in a manner that is appropriate and understandable. The Results demonstrate that the binding proteins evaluated are sufficient to enable highly-multiplexed serology in non-human animals.

Discussion and Conclusions

The Results are discussed in the context of the experimental design and the Conclusion is appropriate. The strengths of the assay (affordable, fast) are well described, as well as the limitations and future directions of the work.

Overall:

The authors have presented a very timely and well-written study aimed at evaluating the utility of IgG-binding proteins A and G, alone or in combination (pA, pG, pAG) in developing a multiplex immunoassay in non-human animals to better understand the distribution and prevalence of viruses that could potentially infect humans.

Points needing clarification:

Comments and suggestions for authors have been made on the draft manuscript as follows:

1. Second to last line of Abstract: Please spell out Fluorescence-Linked Immunosorbent Assay (FLISA) on first use.
2. Second line of Introduction: It would greatly benefit the authors to add the monkeypox outbreak to the list of striking examples since it is ongoing.
3. The Aim or Objectives of the study are not clear and should be clarified in the Abstract. The third sentence of the Discussion "Here, we evaluate the utility of several commercial IgG-binding proteins for enabling highly-multiplexed serology in a wide variety of mammalian species and we describe a simple, competitive FLISA assay that can be used as an initial screen to help identify the best capture protein to use for any mammalian species of interest" could be used as the Aim of the study. Earlier, the authors used "demonstrate" but to clarify the Aim, "evaluating the utility of these IgG binding proteins..." would be a better option.
4. Although the authors indicated that this is a semi-quantitative assay that can be used as a potential screening test, they did not address the sensitivity and specificity of the multiplex immunoassay. In the clinical setting, parameters such as sensitivity, specificity, positive predictive and negative predictive values, and accuracy are measured. Those parameters are lacking here. Please explain.

Reviewer #2 (Public repository details (Required)):

NGS as read-out but did not see accession numbers for SRA or BioProject for NCBI depositing. Needs this.

Reviewer #2 (Comments for the Author):

This manuscript describes use of protein A and protein G to perform highly multiplexed serology in 25 non-human mammals along with candidate validation and discovery work using the multiplexed serology. The capacity of protein A and protein G to work in non-human mammals has been recognized for a long time, though has mostly been limited to model species. This paper tests a broad range of mammals that are relevant for zoonotic viral discovery and serological screening efforts as the authors note. It is also helpful to have a systematic study of rough affinities and best reagents in the context of highly multiplexed serology rather than, say, a monoclonal pulldown. A weakness of the study is the lack of formal determination of K_d's for binding of Fc and limitation of the study to protein A and protein G as capture reagents. The study also is a bit of an odd mishmash of data from the FLISA by mammal and control infection data to the discovery pulldown data in Figure 5, which is intriguing but perhaps not a complete story without the confirmation of the agent, though this could be considered beyond the scope of this paper, which is this reviewer's opinion.

-Authors should specifically note why linear regression used throughout the paper instead of 4PL regression for FLISA dilution data.

It would be nice to see plots such as Figure 5D in supplementary figures for any highly multiplexed pulldown that is performed throughout the paper (this includes Figure 4 pulldowns) to inform pure discovery efforts in any pulldown that is performed. The genome plots are fantastic, but it can somewhat hide the noise of the data if not all the peptides are plotted somewhere.

I did not see any accession numbers or a BioProject for sequencing reads.

No minor edits, the paper is well-written.

Staff Comments:

Preparing Revision Guidelines

Please return the manuscript within 60 days; if you cannot complete the modification within this time period, please contact me. If you do not wish to modify the manuscript and prefer to submit it to another journal, please notify me of your decision immediately so that the manuscript may be formally withdrawn from consideration by Microbiology Spectrum.

Highly-multiplexed serology for non-human mammals

Alexa Schuettenberg¹, Alejandra Piña¹, Morgan Metrailler¹, Ronald Guillermo Peláez-Sánchez², Piedad Agudelo-Flórez², Juan Álvaro Lopez³, Luke Ryle¹, Fernando P. Monroy⁴, John A. Altin⁵, Jason T. Ladner^{1,4*}

1 The Pathogen and Microbiome Institute, Northern Arizona University, Flagstaff, AZ, USA

2 Basic Science Research Group, Graduate School—CES University, Medellín 50021, Colombia

3 Microbiology School, Primary Immunodeficiencies Group, University of Antioquia, Medellín 50010, Colombia

4 Department of Biological Sciences, Northern Arizona University, Flagstaff, AZ, USA

5 The Translational Genomics Research Institute (TGen), Phoenix and Flagstaff, AZ, USA

*Correspondance: jason.ladner@nau.edu

Abstract

Emerging infectious diseases represent a serious and ongoing threat to humans. Most emerging viruses are maintained in stable relationships with other species of animals, and emergence within the human population results from cross-species transmission. Therefore, if we want to be prepared for the next emerging virus, we need to broadly characterize the diversity and ecology of viruses currently infecting other animals (i.e., the animal virosphere). High-throughput metagenomic sequencing has accelerated the pace of virus discovery. However, molecular assays can only detect active viral infections and only if virus is present within the sampled fluid or tissue at the time of collection. In contrast, serological assays measure long-lived antibody responses to infections, which can be detected within the blood, regardless of the infected tissues. Therefore, serological assays can provide a complementary approach to understanding the circulation of viruses, and while serological assays have historically been limited in scope, recent advancements allow 1000s to 100,000s of antigens to be assessed simultaneously using <1 µl of blood (i.e., highly-multiplexed serology). Application of highly-multiplexed serology for characterization of the animal virosphere is dependent on the availability of reagents that can be used to capture or label antibodies of interest. Here, we demonstrate the potential for commercial immunoglobulin-binding proteins (protein A and protein G) to enable highly-multiplexed serology in 25 species of non-human mammals and we describe a competitive FLISA assay that can be used as an initial screen for choosing the most appropriate capture protein for a given host species.

Importance

Antibodies are generated in response to infections with viruses and other pathogens and they help protect against future exposures. Mature antibodies are long-lived, highly specific, and can bind to their protein targets with high affinity. Thus, antibodies can also provide information about an individual's history of viral exposures, which has important applications in understanding the epidemiology and etiology of disease. In recent years, there have been large advances in the available methods for broadly characterizing antibody binding profiles, but thus far, these have primarily been utilized only with human

samples. Here, we demonstrate that commercial antibody-binding reagents can facilitate modern antibody assays for a wide variety of mammalian species, and we describe a cheap and fast approach for choosing the best reagent for each animal species. By studying antibody-binding profiles in captive and wild animals, we can better understand the distribution and prevalence of viruses that could spillover into humans.

Introduction

Emerging and re-emerging infectious diseases (especially those caused by viruses) represent a serious and ongoing threat to the human population. The last couple decades have offered many striking examples of this (e.g., Ebola virus disease (1), Hendra virus disease (2), Middle East respiratory syndrome (3), coronavirus disease 2019 (4)), and both the frequency and impact of emerging infectious diseases are likely to increase in the future due to a variety of factors, including globalization, increases in human population density, and the varied impacts of climate change (5–7). However, we do not know which viruses pose the greatest future risk, which complicates efforts aimed at prevention and mitigation.

One thing we do know is that the vast majority of emerging viruses are maintained in stable relationships with other species of animals, and emergence within the human population results from cross-species transmission facilitated by close contact between humans and animals (8). Therefore, if we want to be prepared for the next emerging virus, we need to broadly characterize the diversity and ecology of the viruses currently infecting other animals (i.e., the animal virosphere) (9). High-throughput metagenomic sequencing has accelerated the pace of virus discovery by enabling deep and broad characterization of nucleic acids (10). However, molecular assays can only detect active (or latent) viral infections and only if virus is present within the sampled fluid or tissue at the time of collection. Therefore, in the context of understanding the animal virosphere, molecular surveillance is akin to searching for a needle in a haystack.

In contrast, serological assays measure long-lived antibody responses to infections, which can be detected within the blood, regardless of the infected tissues. Therefore, serological assays can provide a complementary approach to understanding the circulation of viruses within captive and wild animal populations, and while serological assays have historically been limited in scope (i.e., measuring antibody reactivity against a single antigen at a time), recent advancements now allow 1000s to 100,000s of antigens to be assessed simultaneously using <1 μ l of blood (11, 12). These approaches for highly-multiplexed serology have enabled virome-wide characterization of exposure histories in humans (11), and they also offer the potential for deconvoluting cross-reactive antibody responses (12). However, to our knowledge, highly-multiplexed serology has not been used to broadly assess antiviral antibody reactivity in non-human animals.

Several approaches for highly-multiplexed serology have been described (e.g., PepSeq (12, 13), PhIP-Seq (14), peptide arrays (15)), but they all depend on the availability of reagents that can be used to capture or label antibodies of interest. For example, in both PepSeq and PhIP-Seq, magnetic bead-bound proteins are used to capture antibodies of interest. This step is critical, as it enriches antibody-bound antigens, and it is this enrichment that is used to quantify binding. Similarly, peptide arrays also require proteins that can bind to conserved regions on the antibodies of interest; in this case, these proteins are used to fluorescently label antibodies that have bound to antigens printed on a solid surface. Therefore, the primary limitation for the application of highly-multiplexed serology to non-human animals is the availability of appropriate antibody binding proteins.

The most common antibody (or immunoglobulin, Ig) binding proteins used for both PepSeq and PhIP-Seq bind to human IgG isotype antibodies using IgG-binding domains that were derived from bacterial

proteins. These proteins play a critical role in immune evasion during bacterial infections, and they have been shown to bind to IgG from a variety of mammalian species, including humans (16, 17). Because of this characteristic, several of these bacterial IgG-binding domains have been adapted for use in molecular biology. This includes domains from the Staphylococcal protein A and the Streptococcal protein G, both of which are available commercially, on their own (pA, pG) and in combination (pAG).

In order to evaluate the potential for these commercial immunoglobulin-binding proteins to enable highly-multiplexed serology in various species of mammals, we first developed a competitive fluorescence-linked immunosorbent assay (FLISA) that is able to semi-quantitatively estimate binding affinity directly from complex, antibody-containing samples like serum and plasma. We then used PepSeq to demonstrate the utility of these commercial proteins for running highly-multiplexed serology assays on a wide variety of mammal species, and the utility of the FLISA assay as an initial screen by comparing relative FLISA binding affinity to PepSeq-based quantitative measures of antigen enrichment.

Materials and Methods

Samples

In total, our study included 82 samples from 25 species of non-human animals (including mule, a sterile hybrid), as well as four human samples as controls. Samples were obtained from a variety of sources, including BEI resources, commercial companies and both academic and non-profit organizations (for details see S1 Table). For collection of the Northern quoll sample (BQ), all research methodologies were approved by the University of Queensland animal ethics committee (SBS/541/ 134 12/ANINDILYAKWA/MYBRAINS) and were conducted under a permit provided by the Northern Territory Parks and Wildlife Commission (permit number: 47603) and with the permission of the Traditional Owners of Groote Eylandt, the Anindilyakwa peoples. For the collection of the arctic ground squirrel sample (BAGS), all procedures were approved by the University of Alaska Fairbanks Institutional Animal Care and Use Committee (IACUC #340270 and #864841) and followed the guidelines established by the American Society of Mammalogists (18). Several of the cat and dog samples (713629, 713635, 713677, 713661, 713807, 713761) were collected as described in Yaglom et al. (19) with approval of the Translational Genomics Research Institute's (TGen) Animal Care and Use Committee (IACUC, #20163). Additional dog samples (3226 – 3231) were collected under a protocol approved by the University of Georgia IACUC (#A2017 05-014-Y1-A0). Two of the non-human primate samples (SNPRC-025 and SNPRC-027) were collected from animals involved in studies approved by the Texas Biomedical Research Institute IACUC. Three of the human samples (VW-128, VW-139, VW-229) were collected at Valleywise Health in Phoenix, AZ with approval of the Valleywise Health Institutional Review Board and the Institutional Review Board of Northern Arizona University (#1545420). The mouse serum (MNAU-sterile) was collected under a protocol approved by the IACUC at Northern Arizona University (IACUC approval number 16-008). The bat sera (ZN-165 – ZN-205) were collected with permission from the Autoridad de Licencias Ambientales (ANLA) within the Colombian Ministry of the Environment (ANLA #0790, 7/18/2014) (20).

FLISA assay

To measure the affinity of each capture protein for antibodies from non-human mammal species, we designed a competitive FLISA assay (Fig. 1A), which utilized a known strong binder (human IgG) as a reporter. For this assay, we pre-coated plates with the capture protein of interest, which included recombinant pA, pG and pAG (Pierce/ThermoFisher). For pA and pG, we used both commercially-prepared (Pierce/ThermoFisher) and in-house coated plates; all pAG coated plates were prepared in-house. For the in-house preparations, we added 300 ng of capture protein (diluted in 0.05 M carbonate-bicarbonate coating buffer, pH 9.6) to black MaxiSorp-treated 96-well immuno plates (ThermoFisher) and then incubated the plates at 4°C for 16 hours. Following the incubation, we washed the wells three times with phosphate-buffered saline (pH 7.4) containing 0.05% tween-20 (PBST). We then blocked the plates by adding 150 μ L of SuperBlock T20 (ThermoFisher) to each well followed by a 1 hour incubation at 37°C.

We then added dilutions of antibody-containing samples to the pre-coated plates and incubated for 2 hours at room temperature followed by three washes, each with 150 μ L of PBST, to remove unbound antibodies. Specifically, we added 100 μ L of sample diluted in PBST and assayed each sample at five serial dilutions. For experimental samples (e.g., serum/plasma), we began each series with a 1:100 dilution of the sample, followed by additional 1:3 serial dilutions. We included chicken serum (BEI Resources, NR-3132) on each plate as a negative control; all of the capture proteins we tested have been reported to have no binding affinity for chicken immunoglobulin (16). We also included purified, normal human IgG (ProSci #5503) on each plate as a positive control; all three capture proteins tested have high affinity for human IgG (16, 21). For each positive control dilution series, we initially diluted the human IgG to 16 μ g/mL (pG) or 32 μ g/mL (pA and pAG) and then generated four additional 1:2 serial dilutions. Finally, we added 100 μ L of FITC-conjugated human IgG (9 ng/ μ L, diluted in PBST) to each well, incubated for 1 hour at room temperature and read the fluorescence at 485/528 nm (excitation/emission) after shaking on medium for 5 minutes using a BioTek Synergy HT microplate reader.

Each sample was assayed in duplicate within a single plate and relative affinity between the immobilized capture protein and sample antibodies was quantified as a “positive slope ratio” (PSR) using a custom python script (https://github.com/jtladner/Scripts/blob/master/FLISA/analyzeCompFLISA_dilutionFactors.py). PSR is a measure of the relative affinity of sample antibodies compared to normal human IgG and was calculated by comparing the rate of change in fluorescence with sample dilution between each sample and the positive control dilution series run on the same plate. Specifically, the rate of change for each sample was calculated using the three consecutive dilutions that resulted in the greatest rate of change (Fig. 1B). Our goal was to capture the portion of the curve in which there is a linear relationship between the change in sample concentration and the change in measured fluorescence. Samples that exhibited poor linear correlations and/or low maximum fluorescence (compared to the negative control) were rerun after adjusting the dilution series appropriately. If we observed a poor positive control linear correlation or a significant negative control linear correlation, the entire plate was rerun.

PepSeq Assay

PepSeq technology enables the production of diverse, fully-defined libraries of DNA-barcoded peptides, which can be used to simultaneously assess antibody binding to 10,000s – 100,000s of antigens. All PepSeq assays were performed following the protocol detailed in Ladner et al. (12) and Henson et al. (13). In brief, 0.1 pmol of PepSeq library (5 μ l) was added to a diluted sample (0.5 μ l of sample in 4.5 μ l of SuperBlock) and incubated overnight. The binding reaction was applied to 10 μ l of pre-washed pA- or pG-bearing magnetic beads and incubated for ≥ 15 min to capture antibodies within the sample. The beads were then washed, bound PepSeq probes were eluted by incubation at 95°C for 5 minutes, and the DNA portions of the PepSeq probes were amplified and prepared for Illumina sequencing using barcoded DNA oligos (12). Following PCR cleanup, products were pooled across assays, quantified and sequenced using an Illumina NextSeq instrument (75- or 150- cycle kit). Two slightly different washing procedures were employed for different assay plates, one using a hand-held magnetic bead extractor and the other involving manual pipetting. However, for all direct comparisons between capture proteins for the same sample, we utilized a consistent plate washing procedure. We also ran a limited number of PepSeq assays that combined IgG-binding domains of pA and pG within a single assay. We tested two approaches for this: 1) using a combination pA- and pG-bearing magnetic beads (10 μ l of each were added to each assay, 20 μ l of beads in total) or 2) using magnetic beads bearing recombinant pAG. For the pA + pG combination, PepSeq assays were run exactly as described above. We tested two different types of commercial beads containing recombinant pAG: 1 μ m magnetic beads (Pierce #88802) and 10-40 μ m agarose magnetic beads (Pierce #78609). Normalizing for differences in binding capacity, we added 5 μ L of 1 μ m non-agarose and 2.5 μ L of 10-40 μ m agarose beads to each assay. Per manufacturer recommendations, both types of pAG beads were incubated with each sample for 1 hour prior to washing.

In this study, we utilized two different PepSeq assays: our 244,000 peptide human virome version 1 assay (HV1) and our 99,971 peptide pan-coronavirus assay (PCV). HV1 was designed to broadly cover potential linear epitopes from several hundred viruses known to infect humans and is described in detail in Ladner et al. (12). PCV was designed to broadly cover potential linear epitopes from all sequenced coronaviruses (including Alpha-, Beta-, Delta- and Gammacoronavirus genera) regardless of host species (see S1 File for details). To ensure consistency, all sample:library:capture protein combinations were assayed in duplicate.

PepSeq Analysis

PepSIRF v1.4.0 was used to analyze the sequencing data from the PepSeq assays (22). First, the data were demultiplexed and reads were assigned to peptide antigens using the *demux* module of PepSIRF with up to one mismatch within each barcode (8-12 nt in length) and up to three mismatches in each DNA tag (40 nt in length). A truncated, 40 nt DNA tag (full length = 90 nt) was used to enable sequencing with a 75-cycle kit, and we excluded 70 and 351 peptides from consideration for the HV1 and PCV libraries, respectively, as the first 40 nt of these DNA tags are not unique within these libraries.

Enrichment Z scores were then calculated using the *diffEnrich* pipeline as implemented in the *autopepsirf* QIIME 2 plugin (<https://ladnerlab.github.io/pepsirf-q2-plugin-docs/>) (23, 24). This pipeline automates the analysis protocol described in Ladner et al. (12) and Henson et al. (13). Briefly, this included

normalization to control for differential sequencing depth across samples (*norm* PepSIRF module, *col_sum* method), normalization against buffer-only negative controls (*norm* PepSIRF module, *diff* method) and calculation of Z scores using a 95% highest density interval with peptide bins that were based on 6 – 14 buffer only negative controls (*zscore* PepSIRF module). There was no discernable difference between the buffer-only controls run with pA- and pG-bearing beads from assays with the same PepSeq library. Therefore, for each library, controls of both types were considered together for the formation of bins (*bin* PepSIRF module). Likewise, for the PepSeq comparisons of pA and pG with pAG, buffer-only controls run with all three capture proteins were used to generate the bins.

To ensure consistency in signal across replicates, Z score correlations were visualized using the *repScatters* module in the *ps-plot* QIIME 2 plugin

(<https://ladnerlab.github.io/pepsirf-q2-plugin-docs/plugins/ps-plot/repScatters/>) and manually inspected.

For sample:library:capture protein combinations with poor Z score correlations, we either reran the relevant assays or omitted those assays from the presented data. To identify enriched peptides, we chose thresholds that minimized false positives when analyzing pairs of buffer-only negative controls that were not considered in the generation of bins or normalization. For PCV assays, we used a Z score threshold of 7.5 and for HV1 assays, we used a Z score threshold of 10 in combination with a column sum normalized read count (i.e., reads per million) threshold of 2. These thresholds resulted in an average of 0 and 0.19 enriched peptide calls in 6 and 21 pseudoreplicate pairs of buffer-only negative controls for PCV and HV1, respectively.

To compare assays conducted with the same sample and library but different IgG capture proteins, we calculated average Z scores for each assay across the union of enriched peptides from all assays being compared. Higher Z scores indicate greater enrichment within the assay. Therefore, we expect to see higher average Z scores for assays run with capture proteins that bind substantially better to antibodies from a given species. MAFFT v7.490 was used to align enriched peptides to full length reference proteins (with “--keeplength” and “--addfragments” options) and to align homologous reference proteins from different viral species (default conditions). For animals immunized with specific viral proteins (e.g., influenza A hemagglutinin), NCBI blastp v2.12.0+ (25) was used to compare all assay peptides against a representative sequence of the appropriate immunogen and any peptide with an e-value $\leq 1e-15$ was considered a match. In cases where a specific protein was not specified, we assumed that whole and/or infectious viral particles were used for immunization and therefore, all peptides designed from the relevant virus species were considered to be matches.

Results

FLISA-based estimates of binding affinity

In total, we used our FLISA assay to evaluate the efficiency of pA, pG and pAG capture proteins across 66 samples from 26 mammalian species (including humans). Each sample was run against 1-3 capture proteins (2.9 capture proteins tested on average per sample) and at least one sample per species was run against all three capture proteins (S1 Table). Positive slope ratios (PSRs), a measure of binding efficiency, ranged between 0.019 – 0.962, 0.004 – 1.492, and 0.064 – 0.908, for pA, pG and pAG, respectively, and PSR values were generally consistent across different samples from the same species (Fig. 2). When considering PSRs from all species for which multiple samples were run against the same capture protein

(56 data points across 18 species for each capture protein), we observed PSR standard deviations of 0.047, 0.109, and 0.064 for pA, pG and pAG, respectively.

To assess our ability to measure relative binding affinity directly from complex sample types (e.g., serum, plasma, ascitic fluid), we compared our FLISA PSR values to published measures of relative affinity using purified IgG (11 mammal species; Fig. 3A, S1 Table). For all three capture proteins, we observed significant correlations between our species-level average PSR values and published IgG-specific inhibition data from Eliasson et al. (16) (Pearson correlation p-values: 0.004, 0.003 and 0.022 for pA, pG and pAG, respectively). As expected, the observed correlations were in the negative direction (Fig. 3), reflecting the fact that PSR is positively correlated with relative binding affinity, while the amount of target species IgG needed for 50% inhibition of rabbit IgG (16) is negatively correlated with relative binding affinity.

All of the species tested exhibited average PSR >0.27 for at least one of the capture proteins and $\sim 81\%$ (21/26) exhibited at least one average PSR >0.54 . The lowest maximum PSR values, across the three capture proteins, were for the four tested species of muroid rodents (hamster, deer mouse, mouse and rat: 0.273 – 0.454), followed by the one tested marsupial (Northern quoll: 0.463). There was no correlation between species-level PSR values for pA and pG (Pearson correlation p-value = 0.71), but PSR values did exhibit some correlation with phylogeny (Fig. 2, S1 Fig.). Specifically, species that were closely related phylogenetically tended to exhibit similar PSR values. For example, all members of the Carnivora order (cat, dog and ferret) exhibited high PSR for pA (0.61 – 0.81; $n=9$), but low PSR for pG (0.004 – 0.26; $n=10$). In contrast, all members of the Bovidae (cow, goat, sheep) and Equidae (donkey, mule) families exhibited low PSR for pA (0.02 – 0.36; $n=15$), but high PSR for pG (0.48 – 1.04; $n=14$). All of the bat species tested exhibited moderately high PSR for both pA (0.47 – 0.67, $n=12$) and pG, however, the three species in the Phyllostomidae family exhibited markedly higher pG PSR (0.82 – 1.5, with one outlier at 0.25; $n=7$) than species in the Molossidae and Vespertilionidae families (0.32 – 0.41; $n=5$).

Average PSR values for pAG were generally intermediate between the PSRs for pA and pG (S2 Fig.). For 68% of the tested non-human mammal species (17/25), the average PSR for pAG was intermediate between the average PSRs for pA and pG. For another 28% (7/25), the average PSR for pAG was lower than the average PSR for both pA and pG.

Highly-multiplexed serology using PepSeq

In order to evaluate the potential for pA and pG to enable highly-multiplexed serology for different species of mammals, and to determine whether PSR is a good predictor of capture protein performance, we assayed 1) 65 samples (from 24 species, including humans) with both pA and pG using at least one of our PepSeq libraries and 2) an additional 11 samples (from 8 species) using a single capture protein (pA or pG) (see S1 Table for details). In total, we conducted at least one PepSeq assay for all of the non-human mammal species tested with our FLISA assay ($n=25$), and we detected ≥ 1 enriched peptide for at least one sample/library combination for every species.

To determine whether FLISA PSR is a good predictor of capture protein performance, we compared PSR values to PepSeq Z scores, which provide a quantitative measure of enrichment for each peptide antigen.

For this comparison, we included 61 sample/library combinations that were assayed with both pA and pG (including samples from 22 species) and for which we observed ≥ 5 enriched peptides (see S1 Table for details). Overall, we observed a significant positive correlation between relative capture protein binding affinity (pA - pG) as measured using our FLISA (average species-level PSR difference) and PepSeq (average sample-level Z score difference) assays (Pearson correlation coefficient = 0.525, p-value = $1.38e-5$) (Fig. 3B). There were, however, some notable outliers to this overall trend. For the rat samples (n=2), we estimated somewhat higher binding affinity to pG with our FLISA, but greater PepSeq enrichment with pA. Similarly, we estimated much higher pG affinity for the great fruit-eating bat (*Artibeus lituratus*), but saw slightly higher PepSeq enrichment with pA (n=1,2 for FLISA and PepSeq, respectively).

We also assayed several samples using combinations of the IgG-binding domains from pA and pG. Specifically, we tested two approaches: 1) a commercially available, recombinant protein (pAG) that consists of fused IgG-binding domains from both pA and pG (S3A Fig.) and 2) a mixture of individual pA and pG bearing magnetic beads (S3B Fig.). In some cases, enrichment signal was comparable between assays that used a combination of the IgG-binding domains and those that used just pA or pG (e.g., pA vs. pAG agarose for NR-19260, S3A Fig.). However, in other cases, the enrichment signal was substantially higher using either pA or pG in isolation (e.g., pG vs. pAG for NR-3120, S3A Fig.).

Of the 76 samples assayed using PepSeq, 29 came from non-human mammals that had been experimentally immunized with a viral antigen that is covered by peptides in one or both of our PepSeq libraries. This included immunogens of varying complexity (e.g., full viral particles and individual proteins) and derived from 15 different virus species (see S1 Table for details). In $\sim 86\%$ of these samples (25/29), we detected at least one enriched peptide from the known immunogen, with an average of ~ 28 enriched peptides per immunogen when considering only the library/capture protein combination with the greatest number of enriched peptides for each sample. To finely map epitopes and examine the cross-reactivity of antibodies elicited through immunization, we aligned homologous protein sequences from several viral congeners contained within our HV1 library and examined the distribution of reactive peptides along the alignments (Fig. 4A-B). NR-9404 is goat serum collected following immunization with the E1-E2 glycoprotein of Venezuelan equine encephalitis virus (VEEV). Our assays with HV1 identified 44 enriched VEEV peptides, which included both previously characterized and novel antibody epitopes, as well antibody reactivities that were specific to VEEV and those that appeared to exhibit cross-reactivity with peptides from 1-8 other virus species from the *Alphavirus* genus (Fig. 4A). Similarly, NR-9676 is deer mouse serum collected following immunization with the nucleocapsid protein of Sin Nombre virus (SNV). For NR-9676, we detected antibody reactivity against nine SNV peptides from the nucleocapsid, which together represented ≥ 6 distinct epitopes, only one of which was specific to SNV (i.e., antibodies only recognized SNV peptides at that epitope, Fig. 4B). Additionally, five of the HV1 assayed samples were from goats and sheep that had been experimentally immunized with hemagglutinin protein from different subtypes of influenza A virus (H1, H2, H5, H9 and H12). For each of these samples, we used our PepSeq data to calculate a relative enrichment score for hemagglutinin subtypes 1-14, and in each case, the highest relative enrichment score corresponded with the subtype used for immunization (Fig. 4C).

In our PepSeq assays, we also detected strong signals of antibody reactivity (≥ 5 enriched peptides) against many other viruses, beyond those that were documented as known immunogens, and most of these reactivities are consistent with known host-virus associations. For example, we observed antibodies against erbovirus A peptides in samples from both a donkey and a mule (32 and 13 enriched peptides, respectively). Our analysis identified epitopes within the P1, P2, and P3 regions of the proteome, most of which were not previously reported within the Immune Epitope Database (IEDB), and many of these epitopes were shared between the two samples (Fig. 5A). We also observed antibody reactivity against Betacoronavirus 1 peptides in samples from two Bovidae species (cow, goat) and two Equidae species (donkey, mule) with no documented history of infection/immunization (6 – 12 enriched peptides per sample). Strikingly, we observed strong congruence between the antibody epitopes observed in these samples and those observed in samples from animals with a documented history of immunization (Fig. 5B). Similarly, we observed antibody reactivity against 1) Aichivirus A in two dogs (7 – 8 enriched peptides), three goats (15 – 21 enriched peptides) and a hamster (10 peptides), 2) mammalian orthoreovirus in three goats (7 – 10 enriched peptides) and one bat (5 enriched peptides), 3) influenza A virus in a donkey and a mule (5 – 8 enriched peptides) and 4) all eight species in the *Enterovirus* genus (enterovirus A-E and rhinovirus A-C) in three goats (107 – 174 enriched peptides summed across the eight virus species).

Finally, even in the absence of a well-established link between host and virus, we were able to generate strong evidence for potential past exposures based on 1) the enrichment of multiple peptides overlapping the same epitope, 2) the enrichment of multiple epitopes from the same virus species, genus or family and/or 3) consistent patterns of enrichment across multiple individuals from the same host group. For example, we observed antibody reactivity against peptides designed from Pestivirus A in several of the bat sera we tested (Fig. 5C-D). In total, we observed enriched Pestivirus A peptides in our assays of sera from eight different bats, representing six different species (1 – 7 peptides per individual). These peptides clustered within four different proteins and, in total, represented 7 epitope regions (1 – 2 epitopes per individual), three of which were recurrent across multiple individuals (Fig. 5C).

Notably, three of the assayed samples consistently (across replicates, capture proteins and PepSeq libraries) resulted in high numbers of enriched peptides compared to the other samples in this study. This included one plasma sample from a baboon (SNPRC-025: pA/HV1 = 821 enriched peptides, pG/HV1 = 939), one serum sample from a ferret (NR-19264: pA/HV1: 1308, pA/PCV = 421) and one ascitic fluid sample from a mouse (NR-48961: pA/HV1 = 2965, pG/HV1 = 1571, pG/PCV = 480). These enriched peptide counts are 31–152x higher than the median values we observed for non-human samples with the corresponding PepSeq libraries and 1.9–7.8x higher than the next highest enriched peptide count for a non-human sample (excluding the counts for these three samples). Given that the virus exposure histories for these samples are incompletely documented, we cannot be certain whether all of the enriched peptides are reflective of true exposure to the viruses targeted in our assays, or whether there could be another reason for the high level of reactivity, such as the presence of highly cross-reactive antibodies. Therefore, out of an abundance of caution, we have not included these three samples in our virus-level descriptions of antibody reactivity.

Discussion

Approaches for highly-multiplexed serology (e.g., PepSeq, PhIP-Seq, peptide arrays) enable the simultaneous assessment of antibody binding to 10,000s – 100,000s of peptide antigens from <1 μ l of blood (11, 12). And because antibodies can serve as long-lived biomarkers of past infections, we can use highly-multiplexed serology to better understand the diversity and ecology of the viruses that infect non-human animals and which, therefore, may have the potential to spill over into the human population. Here, we evaluate the utility of several commercial IgG-binding proteins for enabling highly-multiplexed serology in a wide variety of mammalian species and we describe a simple, competitive FLISA assay that can be used as an initial screen to help identify the best capture protein to use for any mammalian species of interest.

Recombinant versions of two bacterial proteins – Staphylococcal protein A (pA) and Streptococcal protein G (pG) – have been developed into commercial reagents and are widely used for immunological applications, including the immunoprecipitation of antibodies to facilitate liquid-phase approaches to highly-multiplexed serology (e.g., PepSeq, PhIP-Seq) (12, 14). The immunoglobulin-binding domains of pA and pG have both been shown to bind to IgG from a variety of mammalian species, but often with very different affinities (16, 17), and to our knowledge, our study represents the first demonstration that this binding is sufficient to enable highly-multiplexed serology in non-human animals.

Using the PepSeq platform (11, 12, 26), we demonstrate that both pA and pG have the potential to broadly enable highly-multiplexed serology in mammalian hosts, but also that assay sensitivity is related to the relative affinity with which these proteins bind antibodies from different species. One of our goals was to design a simple assay that could accurately assess the relative affinity of capture proteins for antibodies from a variety of animal species, and we wanted the assay to work directly with serum/plasma (i.e. not require purified antibodies) and to be robust to initial antibody concentration within the sample. To accomplish these goals, we utilized a multiple-dilution competitive FLISA approach (Fig. 1), with the capture protein of interest fixed to the bottom of each well and FITC-labeled human IgG used as a reporter. We chose human IgG as our reporter because pA and pG are both known to exhibit strong binding affinity for human IgG and both have been shown to work well for immunoprecipitation-based highly-multiplexed serology assays with human samples (11, 12, 26). Using this FLISA assay, we characterized samples from 26 mammalian species and demonstrated strong correlations between our FLISA-based affinities and published relative affinities measured from purified IgG (Fig. 3A), as well as quantitative measures of enrichment (Z scores) from our PepSeq assays (Fig. 3B). These results demonstrate the utility of our FLISA assay for 1) estimating the relative affinities of pA and pG for antibodies directly from complex samples (e.g., serum, plasma, ascitic fluid) and 2) gauging the potential for these capture proteins to enable sensitive highly-multiplexed serology assays. It should be noted that our assay is not able to directly determine which isotypes are binding to the capture proteins. However, based on known isotype-binding profiles for these proteins (27, 28) and our strong correlations with measurements from purified IgG, we expect that most of the bound antibodies, across the various mammal species, will be IgG.

Using pA and/or pG we successfully detected antibody-binding to peptide antigens within samples from all 25 of the tested non-human mammalian species, and this included representatives of 25 genera, 17

families and 8 orders. The presence of enriched peptides demonstrated both a) successful immunoprecipitation of antibodies using pA and/or pG and b) antibody binding to peptide antigens present in our libraries. In fact, we observed significant peptide enrichment within PepSeq assays even with species/capture protein combinations that exhibited relatively low affinities in our FLISA assays. For example, we measured PSR values for our hamster serum at 0.11 and 0.27 for pA and pG, respectively, and yet we observed 6 and 72 enriched peptides when assaying this sample against our HV1 PepSeq assay with pA and pG, respectively. However, we generally observed little, if any, enrichment in cases where $PSR < 0.1$ (pA: cow, goat; pG: cat, ferret). Overall, we expect that future PepSeq libraries, designed specifically to cover viruses known to infect the non-human hosts of interest, will result in more enriched peptides and therefore improved estimates of assay performance.

In general, we observed similar patterns of affinity between antibodies and bacterial IgG-binding proteins for phylogenetically closely related mammal species (Fig. 2 and S1). This is consistent with differences in binding affinity being driven by genetic differences between species in the constant regions of the genes that encode the Ig heavy and light chains. This pattern also suggests that relative binding affinity for untested mammal species can be reasonably approximated by comparison to available data from close relatives. Additionally, by comparing relative binding affinities to Ig heavy and light chain amino acid sequences from a variety of species, we may be able to better understand the binding motifs of these bacterial IgG-binding proteins and therefore predict relative affinities even for species for which data for close relatives is unavailable.

Another potential approach for dealing with differences in the relative affinities of pA and pG across mammal species is to combine these two proteins within a single assay. This could be done with recombinant pAG, which combines the IgG-binding sites of both pA and pG within a single construct, or with a mixture of pA and pG. Our initial tests of both approaches showed some promise, with these combinations sometimes able to match the enrichment signal observed with the highest affinity capture protein in isolation (S3 Fig.). However, we observed quite a bit of variability in the relative performance of these combination approaches with different samples/species. More work is needed to understand whether this relative performance is species-specific and whether these approaches can be further optimized. One important difference is the types of beads on which pAG is commercially available. Our standard protocol is optimized for 2.8 μm Dynabeads (Invitrogen), but Dynabeads does not currently offer a pAG option. For pAG, we have tested 1 μm magnetite-coated polymeric beads (Pierce #88802) and 10-40 μm agarose beads (Pierce #78609). It is also possible that there is some interference between antibody binding at the pA and pG IgG-binding domains on the recombinant pAG constructs, which impacts overall binding in these assays. Additionally, in order to ensure that overall binding capacity was not reduced for species with poor affinity to one of the capture proteins, our pA + pG assays included twice the volume of beads compared to our typical assays with pA or pG in isolation. Therefore, even if these assays were able to consistently match the performance of the individual pA or pG assays, they would be more expensive to run.

In addition to detecting expected reactivities against documented immunogens (Fig. 4), we also observed many additional reactivities that are consistent with documented host/virus relationships. This includes several members of the *Picornaviridae* family. For example, Erbovirus A (i.e., Equine rhinitis B virus), which is a member of the *Erbovirus* genus, is known to be a common respiratory pathogen of horses (29).

We observed broad antibody reactivity against Erbovirus A peptides in serum from both a donkey and a mule (Fig. 5A). We also observed antibody reactivity against Aichivirus A peptides in samples from two dogs, three goats, and one hamster. Aichivirus A is a species within the *Kobuvirus* genus and it includes 1) a virus that has been isolated, on multiple occasions, from diarrhoeic dogs (canine kobuvirus) (30, 31) and 2) viruses that have been identified in association with many different species of rodents (murine and rat kobuviruses) (32–34). Although we are not aware of any examples of Aichivirus A isolation from goats, several closely related kobuviruses have been reported in goats, including Aichivirus B (35, 36) and Aichivirus C (36, 37), neither of which were targeted by our PepSeq assays. We also detected enrichment of mammalian orthoreovirus (family *Reoviridae*) peptides in three goats and one bat (*Myotis cf. caucensis*, *Vespertilionidae* family). Mammalian orthoreovirus is known to infect a broad range of mammal species, and this viral species has been identified within both Eurasia and North America in association with several species of bats in the *Vespertilionidae* family (38–40). Within each of our seropositive samples, we observed reactivity against 3–5 of the ten genome segments (collectively we observed antibody reactivity against 8/10 segments).

Betacoronavirus 1, which is a member of the *Coronaviridae* family, includes a common human pathogen (human coronavirus OC43) as well as viruses that commonly infect a variety of non-human animal species (e.g., bovine coronavirus, equine coronavirus). We detected robust antibody reactivity against betacoronavirus 1 (≥ 6 peptides) in five animals with no documented exposures, and the epitopes recognized in these individuals were similar to those observed following experimental immunization (Fig. 5B). Reactive samples were collected from a cow, a goat, a donkey and two mules. To our knowledge, cows and goats are known hosts for bovine coronavirus (41–43), while horses and donkeys are known hosts for equine coronavirus (44–46). However, our data suggest that the host ranges of these viruses may be broader than currently documented. The most commonly recognized betacoronavirus 1 epitope is located in the Spike protein, overlapping the S1/S2 cleavage site (Fig. 5B). Comparing levels of reactivity at this epitope to homologous peptides from several different viruses, we found that antibody reactivity within the donkey and mules was strongest against bovine coronavirus peptides, while reactivity within the cow was strongest against an equine coronavirus peptide (S4 Fig.) (antibodies within the goat sample did not recognize this epitope).

Also of note, we observed antibody reactivity against pestivirus A peptides in eight serum samples collected from bats in Necoclí-Colombia, including representatives of six species, five genera and three families (Fig. 5C–D). In total, this included reactivity against 14 unique peptides, two of which were recognized by antibodies in multiple individual bats, and collectively, these peptides clustered into seven epitopes. Four of these epitopes are within the E1 and E2 envelope glycoproteins (collectively recognized in six samples) and one is within the NS3 protein (collectively recognized in three samples). All of these proteins are known to be immunogenic (47–50). The two remaining epitopes fell within the RNA-dependent RNA polymerase (collectively recognized in three samples), which is one of the most highly conserved proteins across pestivirus species (S5 Fig.). Pestivirus A is one of 11 recognized species in the *Pestivirus* genus and the only one of these species included in our assay (51). We are not aware of any reports of pestivirus A infections of bats; however, two currently unclassified lineages of bat pestiviruses have been documented via high-throughput sequencing from two species (and two families) of bats in China (*Rhinolophus affinis* and *Scotophilus kuhlii*) (52, 53). Our results suggest that the geographic and taxonomic distributions of bat pestiviruses are substantially broader than currently

recognized. Additionally, many of the pestivirus epitopes recognized by bat antibodies in this study are poorly conserved in the bat pestiviruses from China (S5 Fig.), which are distantly related to pestivirus A (51). Therefore, these reactivities may represent infections with distinct bat pestivirus lineages.

Supporting Information

S1 File. Pan-coronavirus PepSeq Library.

S1 Table. Sample metadata.

S1 Figure. Species with close phylogenetic relationships tend to exhibit similar estimated binding affinities to the IgG-binding proteins tested in this study.

S2 Figure. Relative binding affinities measured for protein AG (orange) were often intermediate between affinities measured for protein A (green) and protein G (blue).

S3 Figure. Average *Z* scores from PepSeq assays using protein A and protein G IgG-binding domains as capture proteins, both in isolation and in combination.

S4 Figure. PepSeq enrichment *Z* scores for diverse peptides covering a single epitope region of the betacoronavirus 1 Spike protein.

S5 Figure. Sequence conservation (amino acid level) across 14 PepSeq peptides (x-axis) and 11 species and two unclassified viruses within the *Pestivirus* genus (y-axis).

Data availability statement

All relevant data are within the paper, within Supporting Information files and/or have been deposited in OSF (<https://osf.io/8hygc/>).

Funding

This work was supported by Ministerio de Ciencia Tecnología e Innovación of Colombia (122877757660 to PAF). The funders had no role in study design, data collection and analysis, decision to publish, or preparation of the manuscript.

Competing interests

The authors declare no competing interests.

Acknowledgements

We would like to acknowledge the following people for their assistance in acquiring samples for this project: Loren Buck (NAU), Rachel Caballero (ValleyWise Health), Emily Cope (NAU), Luis Giavedoni

(Trinity University), Sierra Jaramillo (NAU), Mary Mulrow (ValleyWise Health), Lora Nordstrom (ValleyWise Health), Dan Quan (ValleyWise Health), Chandler Roe (NAU), Heather Venkat (ADHS), Guilherme Verocai (UGA), Cory T Williams (CSU), Robbie Wilson (University of Queensland) and Hayley Yaglom (TGen). For detailed acknowledgements of samples obtained from BEI Resources, see S1 Table. This work was supported by the State of Arizona Technology and Research Initiative Fund (TRIF, administered by the Arizona Board of Regents, through Northern Arizona University) and MINCIENCIAS Colombia (Project: 122877757660—Infectious agents in bats: Contribution to the diagnosis of acute febrile syndrome of zoonotic origin in Urabá region [Antioquia-Colombia]) call for projects of Science, Technology and Innovation in Health—2017 (Call 777).

Figures

Figure 1. Competitive FLISA assay provides semi-quantitative measure of the binding affinity between immunoglobulin-binding proteins and sample antibodies. A) Diagram depicting the basic steps of the competitive FLISA assay presented here. Purple shapes represent immunoglobulin capture proteins. Unlabeled sample antibodies are shown in red, while fluorescently-labeled control antibodies (known to have strong affinity to the capture protein) are shown in green. The differently shaped and colored molecules in step 2 represent various other, non-target proteins contained within the complex sample used as input for the assay. B) An example of the results from a single protein A (pA) FLISA plate, with negative (chicken serum) and positive (human IgG) controls shown in shades of orange and experimental samples (mouse ascitic fluid, sheep and rabbit serum) shown in shades of blue. Every sample was run at five dilutions, two replicates per dilution, and “x” represents the highest concentration, which varied among samples. The data points chosen for calculating the slope are outlined in gray and the best fitting line through those data points is plotted. For control samples, the best fit slopes are shown as dotted lines. For the experimental samples, the degree of spacing between line segments is related to the estimated slope. A steeper slope indicates stronger affinity between sample antibodies and pA.

Figure 2. Evolutionarily related species exhibit similar binding profiles. FLISA-based positive slope ratios for protein A (green), protein AG (orange) and protein G (blue). Each point represents a unique sample/capture protein combination, with samples from the same species shown at the same vertical position. Between 1 and 6 samples were assayed per species/capture protein combination. Species are oriented along the y-axis in accordance with the phylogenetic tree shown on the left, which was generated using TimeTree (54) on July 23, 2021. Three substitutions were made when generating the phylogenetic tree because of missing species from the TimeTree database [*Uroderma bilobatum* in place of *Uroderma convexum* (“tent-making bat”), *Myotis nigricans* in place of *Myotis cf. caucensis* (“mouse-eared bat”), *Equus ferus* in place of mule]. See S1 Figure for correlation between evolutionary divergence and the pairwise difference in positive slope ratio. For scientific names of each species, see S1 Table.

Figure 3. FLISA-based positive slope ratio (PSR) from complex samples correlates with relative binding affinity of purified IgG and peptide-level enrichment in highly-multiplexed serology assays. A) Scatterplot comparing our FLISA-based PSR estimates from unfractionated plasma/serum/ascitic fluid (x-axis) with published (16) relative affinity estimates from purified polyclonal IgG. Each point represents a unique species/capture protein combination, with 11 species represented per capture protein (cow, dog, goat, guinea pig, horse/mule, human, mouse, pig, rabbit, rat, sheep). Our FLISA data from mules was compared to the horse results from Eliasson et al. (16). PSR values represent the average across samples, when multiple samples were analyzed from the same species (S1 Table). Lines represent best fit linear regressions generated using the regplot function in the seaborn python module (Pearson correlation p-values: 0.004, 0.02 and 0.002 for protein A, AG and G, respectively). B) Scatterplot comparing the difference (protein A - protein G) in average PSR (x-axis) to the average Z score difference (protein A - protein G) for peptides enriched above a threshold within a PepSeq assay using either capture protein. Each point represents a single sample assayed with both FLISA and PepSeq using both protein A and protein G as capture proteins. Solid line represents the best fit linear regression generated using the regplot function in the seaborn python module with the 95% confidence interval (1000 bootstraps) shown with the gray ribbon (Pearson correlation p-value = 1.38×10^{-5}). Dotted lines are drawn at x=0 and y=0 for reference.

Figure 4. Epitope-resolved antibody binding profiles against experimental immunogens. Distribution of enriched peptides (contained within the HV1 PepSeq library) across alignments of A) E2-E1 glycoproteins of several species in the *Alphavirus* genus from PepSeq assays (pA and pG) of goat serum (NR-9404) collected following experimental immunization with Venezuelan equine encephalitis virus (VEEV), TC-83 (subtype IA/B) glycoprotein and B) nucleocapsid proteins of several species in the *Orthohantavirus* genus from PepSeq assays (pA and pG) of deer mouse serum (NR-9676) collected following experimental immunization with Nucleocapsid protein of the SN77734 strain of Sin Nombre virus (SNV). In (A) and (B), each row represents assay peptides designed from a different virus species and gray rectangles represent regions containing known epitopes from the species used for immunization (SNV) as documented in the Immune Epitope Database (IEDB.org). C) Relative enrichment scores for 14 subtypes of influenza A virus hemagglutinin (H1-H14) calculated using PepSeq Z scores from assays of serum [NR-3148 (goat, H1), NR-4523 (goat, H2), NR-622 (sheep, H9), NR-19222 (goat, H12)] or purified immunoglobulin [NR-49241 (sheep, H5)] from five animals following experimental immunization with a specific subtype of influenza A virus hemagglutinin. Each row represents a single sample/capture protein combination. The hemagglutinin subtype used for immunization, mammal species and capture protein are all indicated in the y-axis labels. Additional abbreviations in (A): chikungunya virus (CHIKV), eastern equine encephalitis virus (EEEV), Highlands J virus (HJV), Mayaro virus (MAYV), Mosso das Pedras virus (MDPV), Ndumu virus (NDUV), Pixuna virus (PIXV), Rio Negro virus (RNV), Ross River virus (RRV), Sindbis virus (SINV), Tonate virus (TONV). Additional

abbreviations in (B): Andes orthohantavirus (ANDV), Anjzorobe orthohantavirus (ANJZV), Black Creek Canal orthohantavirus (BCCV), Choclo orthohantavirus (CHOV), Dobrava-Belgrade orthohantavirus (DOBV), Hantaan orthohantavirus (HTNV), Laguna Negra orthohantavirus (LANV), Puumala orthohantavirus (PUUV), Sangassou orthohantavirus (SANGV), Seoul orthohantavirus (SEOV).

Figure 5. Epitope-resolved antibody binding profiles against viruses not associated with known infections/immunizations. Distribution of enriched peptides (contained within the HV1 PepSeq library) across the A) erbovirus A polyprotein, B) betacoronavirus 1 Spike and Nucleocapsid proteins and C) pestivirus A polyprotein. The y-axis labels indicate the mammal species of origin, sample ID, and for (A) and (B), the capture protein used for the PepSeq assay. For (C), both pA and pG assays were considered for each sample, when available. In (A) and (C), the gray rectangles represent regions containing known epitopes as documented in the Immune Epitope Database (IEDB.org). Green dashed lines delineate labeled protein regions, while the black dotted line in (B) separates hosts with documented betacoronavirus 1 immunizations (“DI”, bottom) from those without documented immunizations (top). D) Scatter plot comparing normalized read counts from HV1 library PepSeq assays of buffer-only negative controls (x-axis, 11 replicates) and a serum sample from a tent-making bat (ZN-168, y-axis, 2 replicates). Each circle represents a unique peptide. Enriched peptides are shown as larger, colored circles, while non-enriched peptides are shown in smaller, gray circles. The seven orange circles represent enriched peptides from pestivirus A [see (C) for locations within the polypeptide], while all other enriched peptides are shown in blue.

References

1. Baize S, Pannetier D, Oestereich L, Rieger T, Koivogui L, Magassouba N 'faly, Soropogui B, Sow MS, Keïta S, De Clerck H, Tiffany A, Dominguez G, Loua M, Traoré A, Kolié M, Malano ER, Heleze E, Bocquin A, Mély S, Raoul H, Caro V, Cadar D, Gabriel M, Pahlmann M, Tappe D, Schmidt-Chanasit J, Impouma B, Diallo AK, Formenty P, Van Herp M, Günther S. 2014. Emergence of Zaire Ebola virus disease in Guinea. *N Engl J Med* 371:1418–1425.
2. Plowright RK, Foley P, Field HE, Dobson AP, Foley JE, Eby P, Daszak P. 2011. Urban habituation, ecological connectivity and epidemic dampening: the emergence of Hendra virus from flying foxes (*Pteropus* spp.). *Proc Biol Sci* 278:3703–3712.
3. Sharif-Yakan A, Kanj SS. 2014. Emergence of MERS-CoV in the Middle East: Origins, Transmission, Treatment, and Perspectives. *PLoS Pathogens*
<https://doi.org/10.1371/journal.ppat.1004457>.
4. Li Q, Guan X, Wu P, Wang X, Zhou L, Tong Y, Ren R, Leung KSM, Lau EHY, Wong JY, Xing X, Xiang N, Wu Y, Li C, Chen Q, Li D, Liu T, Zhao J, Liu M, Tu W, Chen C, Jin L, Yang R, Wang Q, Zhou S, Wang R, Liu H, Luo Y, Liu Y, Shao G, Li H, Tao Z, Yang Y, Deng Z, Liu B, Ma Z, Zhang Y, Shi G, Lam TTY, Wu JT, Gao GF, Cowling BJ, Yang B, Leung GM, Feng Z. 2020. Early Transmission Dynamics in Wuhan, China, of Novel Coronavirus-Infected Pneumonia. *N Engl J Med* 382:1199–1207.
5. Marston HD, Folkers GK, Morens DM, Fauci AS. 2014. Emerging viral diseases: confronting threats with new technologies. *Sci Transl Med* 6:253ps10.
6. Howard CR, Fletcher NF. 2012. Emerging virus diseases: can we ever expect the unexpected? *Emerg Microbes Infect* 1:e46.
7. Patz JA, Daszak P, Tabor GM, Aguirre AA, Pearl M, Epstein J, Wolfe ND, Kilpatrick AM,

- Foufopoulos J, Molyneux D, Bradley DJ, Working Group on Land Use Change and Disease Emergence. 2004. Unhealthy landscapes: Policy recommendations on land use change and infectious disease emergence. *Environ Health Perspect* 112:1092–1098.
8. Woolhouse M, Scott F, Hudson Z, Howey R, Chase-Topping M. 2012. Human viruses: discovery and emergence. *Philosophical Transactions of the Royal Society B: Biological Sciences* <https://doi.org/10.1098/rstb.2011.0354>.
 9. Ladner JT. 2021. Genomic signatures for predicting the zoonotic potential of novel viruses. *PLoS Biol*.
 10. Zhang Y-Z, Shi M, Holmes EC. 2018. Using Metagenomics to Characterize an Expanding Virosphere. *Cell* 172:1168–1172.
 11. Xu GJ, Kula T, Xu Q, Li MZ, Vernon SD, Ndung'u T, Ruxrungtham K, Sanchez J, Brander C, Chung RT, O'Connor KC, Walker B, Larman HB, Elledge SJ. 2015. Viral immunology. Comprehensive serological profiling of human populations using a synthetic human virome. *Science* 348:aaa0698.
 12. Ladner JT, Henson SN, Boyle AS, Engelbrektson AL, Fink ZW, Rahee F, D'ambrozio J, Schaecher KE, Stone M, Dong W, Dadwal S, Yu J, Caligiuri MA, Cieplak P, Bjørås M, Fenstad MH, Nordbø SA, Kainov DE, Muranaka N, Chee MS, Shiryayev SA, Altin JA. 2021. Epitope-resolved profiling of the SARS-CoV-2 antibody response identifies cross-reactivity with endemic human coronaviruses. *Cell Rep Med* 2:100189.
 13. Henson SN, Elko EA, Swiderski P, Liang Y, Engelbrektson A, Piña A, Boyle AS, Fink ZW, Facista S, Martinez V, Rahee F, Brown A, Kelley EJ, Nelson GA, Raspet I, Mead HL, Altin JA, Ladner JT. PepSeq: a fully in-vitro platform for highly-multiplexed serology using customizable DNA-barcoded peptide libraries. *Nat Protoc*.

14. Mohan D, Wansley DL, Sie BM, Noon MS, Baer AN, Laserson U, Larman HB. 2018. PhIP-Seq characterization of serum antibodies using oligonucleotide-encoded peptidomes. *Nat Protoc* 13:1958–1978.
15. Mishra N, Huang X, Joshi S, Guo C, Ng J, Thakkar R, Wu Y, Dong X, Li Q, Pinapati RS, Sullivan E, Caciula A, Tokarz R, Briese T, Lu J, Lipkin WI. 2021. Immunoreactive peptide maps of SARS-CoV-2. *Commun Biol* 4:225.
16. Eliasson M, Olsson A, Palmcrantz E, Wiberg K, Inganäs M, Guss B, Lindberg M, Uhlén M. 1988. Chimeric IgG-binding receptors engineered from staphylococcal protein A and streptococcal protein G. *J Biol Chem* 263:4323–4327.
17. Kelly PJ, Tagwira M, Matthewman L, Mason PR, Wright EP. 1993. Reactions of sera from laboratory, domestic and wild animals in Africa with protein A and a recombinant chimeric protein AG. *Comp Immunol Microbiol Infect Dis* 16:299–305.
18. Sikes RS, Animal Care and Use Committee of the American Society of Mammalogists. 2016. 2016 Guidelines of the American Society of Mammalogists for the use of wild mammals in research and education. *J Mammal* 97:663–688.
19. Yaglom HD, Hecht G, Goedderz A, Jasso-Selles D, Ely JL, Ruberto I, Bowers JR, Engelthaler DM, Venkat H. 2021. Genomic investigation of a household SARS-CoV-2 disease cluster in Arizona involving a cat, dog, and pet owner. *One Health* 13:100333.
20. Monroy FP, Solari S, Lopez JÁ, Agudelo-Flórez P, Sánchez RGP. 2021. High Diversity of *Leptospira* Species Infecting Bats Captured in the Urabá Region (Antioquia-Colombia). *Microorganisms* <https://doi.org/10.3390/microorganisms9091897>.
21. Akerström B, Björck L. 1986. A physicochemical study of protein G, a molecule with unique immunoglobulin G-binding properties. *Journal of Biological Chemistry*

[https://doi.org/10.1016/s0021-9258\(18\)67515-5](https://doi.org/10.1016/s0021-9258(18)67515-5).

22. Fink ZW, Martinez V, Altin J, Ladner JT. 2020. PepSIRF: a flexible and comprehensive tool for the analysis of data from highly-multiplexed DNA-barcoded peptide assays. arXiv preprint arXiv:2007.05050.
23. Bolyen E, Rideout JR, Dillon MR, Bokulich NA, Abnet CC, Al-Ghalith GA, Alexander H, Alm EJ, Arumugam M, Asnicar F, Bai Y, Bisanz JE, Bittinger K, Brejnrod A, Brislawn CJ, Brown CT, Callahan BJ, Caraballo-Rodríguez AM, Chase J, Cope EK, Da Silva R, Diener C, Dorrestein PC, Douglas GM, Durall DM, Duvallet C, Edwardson CF, Ernst M, Estaki M, Fouquier J, Gauglitz JM, Gibbons SM, Gibson DL, Gonzalez A, Gorlick K, Guo J, Hillmann B, Holmes S, Holste H, Huttenhower C, Huttley GA, Janssen S, Jarmusch AK, Jiang L, Kaehler BD, Kang KB, Keefe CR, Keim P, Kelley ST, Knights D, Koester I, Kosciulek T, Kreps J, Langille MGI, Lee J, Ley R, Liu Y-X, Loftfield E, Lozupone C, Maher M, Marotz C, Martin BD, McDonald D, McIver LJ, Melnik AV, Metcalf JL, Morgan SC, Morton JT, Naimey AT, Navas-Molina JA, Nothias LF, Orchanian SB, Pearson T, Peoples SL, Petras D, Preuss ML, Pruesse E, Rasmussen LB, Rivers A, Robeson MS 2nd, Rosenthal P, Segata N, Shaffer M, Shiffer A, Sinha R, Song SJ, Spear JR, Swafford AD, Thompson LR, Torres PJ, Trinh P, Tripathi A, Turnbaugh PJ, Ul-Hasan S, van der Hooft JJJ, Vargas F, Vázquez-Baeza Y, Vogtmann E, von Hippel M, Walters W, Wan Y, Wang M, Warren J, Weber KC, Williamson CHD, Willis AD, Xu ZZ, Zaneveld JR, Zhang Y, Zhu Q, Knight R, Caporaso JG. 2019. Reproducible, interactive, scalable and extensible microbiome data science using QIIME 2. *Nat Biotechnol* 37:852–857.
24. Brown AM, Bolyen E, Rasset I, Altin JA, Ladner JT. 2022. PepSIRF + QIIME 2: software tools for automated, reproducible analysis of highly-multiplexed serology data. arXiv arXiv:2207.11509.
25. Camacho C, Coulouris G, Avagyan V, Ma N, Papadopoulos J, Bealer K, Madden TL. 2009. BLAST : architecture and applications. *BMC Bioinformatics* <https://doi.org/10.1186/1471-2105-10-421>.

26. Vogl T, Klompus S, Leviatan S, Kalka IN, Weinberger A, Wijmenga C, Fu J, Zhernakova A, Weersma RK, Segal E. 2021. Population-wide diversity and stability of serum antibody epitope repertoires against human microbiota. *Nat Med* 27:1442–1450.
27. Björck L, Kronvall G. 1984. Purification and some properties of streptococcal protein G, a novel IgG-binding reagent. *J Immunol* 133:969–974.
28. Goding JW. 1978. Use of staphylococcal protein A as an immunological reagent. *J Immunol Methods* 20:241–253.
29. Horsington J, Lynch SE, Gilkerson JR, Studdert MJ, Hartley CA. 2013. Equine picornaviruses: well known but poorly understood. *Vet Microbiol* 167:78–85.
30. Li L, Pesavento PA, Shan T, Leutenegger CM, Wang C, Delwart E. 2011. Viruses in diarrhoeic dogs include novel kobuviruses and sapoviruses. *J Gen Virol* 92:2534–2541.
31. Kapoor A, Simmonds P, Dubovi EJ, Qaisar N, Henriquez JA, Medina J, Shields S, Lipkin WI. 2011. Characterization of a canine homolog of human Aichivirus. *J Virol* 85:11520–11525.
32. Zhang M, You F, Wu F, He H, Li Q, Chen Q. 2021. Epidemiology and genetic characteristics of murine kobuvirus from faecal samples of *Rattus losea*, *Rattus tanezumi* and *Rattus norvegicus* in southern China. *Journal of General Virology* <https://doi.org/10.1099/jgv.0.001646>.
33. You F-F, Zhang M-Y, He H, He W-Q, Li Y-Z, Chen Q. 2020. Kobuviruses carried by *Rattus norvegicus* in Guangdong, China. *BMC Microbiol* 20:94.
34. Phan TG, Kapusinszky B, Wang C, Rose RK, Lipton HL, Delwart EL. 2011. The fecal viral flora of wild rodents. *PLoS Pathog* 7:e1002218.
35. Di Martino B, Di Profio F, Robetto S, Fruci P, Sarchese V, Palombieri A, Melegari I, Orusa R, Martella V, Marsilio F. 2021. Molecular Survey on Kobuviruses in Domestic and Wild Ungulates

From Northwestern Italian Alps. *Front Vet Sci* 8:679337.

36. Melegari I, Di Profio F, Sarchese V, Martella V, Marsilio F, Di Martino B. 2016. First molecular evidence of kobuviruses in goats in Italy. *Arch Virol* 161:3245–3248.
37. Oem J-K, Lee M-H, Lee K-K, An D-J. 2014. Novel Kobuvirus species identified from black goat with diarrhea. *Vet Microbiol* 172:563–567.
38. Feng KH, Brown JD, Turner GG, Holmes EC, Allison AB. 2022. Unrecognized diversity of mammalian orthoreoviruses in North American bats. *Virology* 571:1–11.
39. Kohl C, Lesnik R, Brinkmann A, Ebinger A, Radonić A, Nitsche A, Mühldorfer K, Wibbelt G, Kurth A. 2012. Isolation and Characterization of Three Mammalian Orthoreoviruses from European Bats. *PLoS ONE* <https://doi.org/10.1371/journal.pone.0043106>.
40. Lelli D, Moreno A, Lavazza A, Bresaola M, Canelli E, Boniotti MB, Cordioli P. 2013. Identification of Mammalian Orthoreovirus Type 3 in Italian Bats. *Zoonoses and Public Health* <https://doi.org/10.1111/zph.12001>.
41. Gumusova O, Yazici Z, Albayrak H, Çakiroglu D. 2007. First report of bovine rotavirus and bovine coronavirus seroprevalance in goats in Turkey. *Veterinarski glasnik* <https://doi.org/10.2298/vetgl0702075g>.
42. Burimuah V, Sylverken A, Owusu M, El-Duah P, Yeboah R, Lamptey J, Frimpong YO, Agbenyega O, Folitse R, Emikpe B, Tasiame W, Owiredu E-W, Oppong S, Antwi C, Adu-Sarkodie Y, Drosten C. 2020. Molecular-based cross-species evaluation of bovine coronavirus infection in cattle, sheep and goats in Ghana. *BMC Vet Res* 16:405.
43. Clark MA. 1993. Bovine coronavirus. *Br Vet J* 149:51–70.
44. Qi P-F, Gao X-Y, Ji J-K, Zhang Y, Yang S-H, Cheng K-H, Cui N, Zhu M-L, Hu T, Dong X, Yan B,

- Wang C-F, Yang H-J, Shi W-F, Zhang W. 2022. Identification of a Recombinant Equine Coronavirus in Donkey, China. *Emerg Microbes Infect* 1–16.
45. Fielding CL, Higgins JK, Higgins JC, McIntosh S, Scott E, Giannitti F, Mete A, Pusterla N. 2015. Disease associated with equine coronavirus infection and high case fatality rate. *J Vet Intern Med* 29:307–310.
 46. Pusterla N, Vin R, Leutenegger C, Mittel LD, Divers TJ. 2016. Equine coronavirus: An emerging enteric virus of adult horses. *Equine Vet Educ* 28:216–223.
 47. Kalaycioglu AT, Russell PH, Howard CR. 2012. The Characterization of the Neutralizing Bovine Viral Diarrhea Virus Monoclonal Antibodies and Antigenic Diversity of E2 Glycoprotein. *Journal of Veterinary Medical Science* <https://doi.org/10.1292/jvms.11-0187>.
 48. Li Y, Jia Y, Wen K, Liu H, Gao M, Ma B, Zhang W, Wang J. 2013. Mapping B-cell linear epitopes of NS3 protein of bovine viral diarrhea virus. *Vet Immunol Immunopathol* 151:331–336.
 49. Deregt D, Dubovi EJ, Jolley ME, Nguyen P, Burton KM, Gilbert SA. 2005. Mapping of two antigenic domains on the NS3 protein of the pestivirus bovine viral diarrhea virus. *Veterinary Microbiology* <https://doi.org/10.1016/j.vetmic.2005.02.010>.
 50. van Rijn PA, Miedema GK, Wensvoort G, van Gennip HG, Moormann RJ. 1994. Antigenic structure of envelope glycoprotein E1 of hog cholera virus. *J Virol* 68:3934–3942.
 51. Smith DB, Meyers G, Bukh J, Gould EA, Monath T, Scott Muerhoff A, Pletnev A, Rico-Hesse R, Stapleton JT, Simmonds P, Becher P. 2017. Proposed revision to the taxonomy of the genus Pestivirus, family Flaviviridae. *Journal of General Virology* <https://doi.org/10.1099/jgv.0.000873>.
 52. Wu Z, Liu B, Du J, Zhang J, Lu L, Zhu G, Han Y, Su H, Yang L, Zhang S, Liu Q, Jin Q. 2018. Discovery of Diverse Rodent and Bat Pestiviruses With Distinct Genomic and Phylogenetic

Characteristics in Several Chinese Provinces. *Front Microbiol* 9:2562.

53. Wu Z, Ren X, Yang L, Hu Y, Yang J, He G, Zhang J, Dong J, Sun L, Du J, Liu L, Xue Y, Wang J, Yang F, Zhang S, Jin Q. 2012. Virome analysis for identification of novel mammalian viruses in bat species from Chinese provinces. *J Virol* 86:10999–11012.
54. Kumar S, Stecher G, Suleski M, Hedges SB. 2017. TimeTree: A Resource for Timelines, Timetrees, and Divergence Times. *Mol Biol Evol* 34:1812–1819.

*Our responses are in blue.

Reviewer #1 (Comments for the Author):

Title: Highly-multiplexed serology for non-human mammals

Review

Abstract, Title, References:

The aim/objectives of the study should be made clear. The authors have "demonstrated" the potential for commercial immunoglobulin-binding proteins, protein A and protein G, to enable highly multiplexed serology in non-human animals using a competitive Fluorescence-Linked Immunosorbent Assay (FLISA). However, the aim of the study, although implicit in the Abstract and Introduction, should be clear. The authors clarified the objective "to evaluate the utility of..." later in the manuscript. That should be mentioned much earlier. What is clear, is the study design, performance, and Results.

As suggested by the reviewer, we have adjusted our phrasing in the last sentence of the abstract, which describes the aim of the study:

"Here, we evaluate the utility of commercial immunoglobulin-binding proteins (protein A and protein G) to enable highly-multiplexed serology in 25 species of non-human mammals and we describe a competitive fluorescence-linked immunosorbent assay (FLISA) assay that can be used as an initial screen for choosing the most appropriate capture protein for a given host species."

The title communicates the intent of the authors clearly and succinctly and is informative and relevant.

The references are relevant, recent, referenced correctly, and appropriate key studies are included. Since the monkeypox outbreak is new and ongoing, the authors could reference that outbreak to strengthen their argument for testing non-human mammals to identify current and emerging pathogens that could potentially impact public health.

As suggested by the reviewer, we have added a reference to the ongoing monkeypox outbreak to our Introduction:

"The last couple decades have offered many striking examples of this [e.g., Ebola virus disease (1), Hendra virus disease (2), Middle East respiratory syndrome (3), coronavirus disease 2019 (4), monkeypox (5)], and..."

"5. Thornhill JP, Barkati S, Walmsley S, Rockstroh J, Antinori A, Harrison LB, Palich R, Nori A, Reeves I, Habibi MS, Apea V, Boesecke C, Vandekerckhove L, Yakubovskiy M, Sendagorta E, Blanco JL, Florence E, Moschese D, Maltez FM, Goorhuis A, Pourcher V, Migaud P, Noe S, Pintado C, Maggi F, Hansen A-BE, Hoffmann C, Lezama JI, Mussini C, Cattelan A, Makofane K,

Tan D, Nozza S, Nemeth J, Klein MB, Orkin CM, SHARE-net Clinical Group. 2022. Monkeypox Virus Infection in Humans across 16 Countries - April-June 2022. N Engl J Med 387:679–691.”

Introduction:

The authors have clearly described the use and benefits of multiplexed serological testing in humans and identified a gap in applying these approaches to non-human subjects.

Given what is already known and the gaps in our knowledge, the research question is not only justified but essential to arresting future epidemics and pandemics, especially those associated with zoonotic infections by identifying antibody or therapeutic targets if these pathogens make the leap from animals to humans like the SARS-CoV-2 coronavirus.

We are glad to hear that the reviewer shares our enthusiasm for this subject area.

Materials and Methods:

This section is clear. The authors have included enough detail in the M&M section that their experimental design can be replicated. All animal and human samples used in the study were IACUC- or IRB-approved by appropriate institutions.

Controls, sampling, statistical analyses, and reporting are well described and appropriate for the study.

Thank you.

Results:

Although the experimental design and statistical approaches are complex, the authors have presented the data in a manner that is appropriate and understandable. The Results demonstrate that the binding proteins evaluated are sufficient to enable highly-multiplexed serology in non-human animals.

Thank you.

Discussion and Conclusions

The Results are discussed in the context of the experimental design and the Conclusion is appropriate. The strengths of the assay (affordable, fast) are well described, as well as the limitations and future directions of the work.

Thank you.

Overall:

The authors have presented a very timely and well-written study aimed at evaluating the utility of IgG-binding proteins A and G, alone or in combination (pA, pG, pAG) in developing a multiplex immunoassay in non-human animals to better understand the distribution and prevalence of viruses that could potentially infect humans.

Thank you.

Points needing clarification:

Comments and suggestions for authors have been made on the draft manuscript as follows:

1. Second to last line of Abstract: Please spell out Fluorescence-Linked Immunosorbent Assay (FLISA) on first use.

We have made this modification in the text, as suggested.

2. Second line of Introduction: It would greatly benefit the authors to add the monkeypox outbreak to the list of striking examples since it is ongoing.

We have added this example, as suggested.

3. The Aim or Objectives of the study are not clear and should be clarified in the Abstract. The third sentence of the Discussion " Here, we evaluate the utility of several commercial IgG-binding proteins for enabling highly-multiplexed serology in a wide variety of mammalian species and we describe a simple, competitive FLISA assay that can be used as an initial screen to help identify the best capture protein to use for any mammalian species of interest" could be used as the Aim of the study. Earlier, the authors used "demonstrate" but to clarify the Aim, "evaluating the utility of these IgG binding proteins..." would be a better option.

We have adjusted our phrasing in the last sentence of the abstract, as suggested.

4. Although the authors indicated that this is a semi-quantitative assay that can be used as a potential screening test, they did not address the sensitivity and specificity of the multiplex immunoassay. In the clinical setting, parameters such as sensitivity, specificity, positive predictive and negative predictive values, and accuracy are measured. Those parameters are lacking here. Please explain.

This comment is well taken, and we agree that this could be a common source of confusion. The term "semi-quantitative" is used in two places in the manuscript: the introduction and the Figure 1 legend. In both places, it is in reference to the FLISA assay that is used to estimate overall binding affinity between protein A/G and serum antibodies. This particular assay would not be directly applicable to a clinical setting because it does not detect antibodies in an antigen-specific manner.

However, the PepSeq assays do provide antigen-specific measures of antibody binding (i.e., enrichment Z scores). These antigen-specific measures of antibody binding (which will be influenced both by the concentration and affinity of the antibodies) could potentially be useful in a clinical context, but this use case is not the focus of our study. Our focus here is simply on demonstrating the feasibility of conducting highly-multiplexed serology assays for non-human mammals. Furthermore, our main goal is for this approach to serve as an exploratory tool that can be used for hypothesis generation and the identification of antigens that could be useful for more targeted assays.

In order to estimate parameters such as sensitivity, specificity, positive predictive and negative predictive values, we would need appropriate cohorts of animals with large numbers of known positive and negative individuals, and there are other (less highly multiplexed) serology approaches that are likely better suited for these types of targeted studies.

We have clarified these points by adding the following paragraph to the Discussion:

“By characterizing several blood samples collected following experimental immunizations, we were able to demonstrate the power of highly-multiplexed serology for carefully dissecting the immune response against specific antigens, including the presence of antibodies that may cross-recognize homologous antigens from related viruses (Fig. 4). By simultaneously assaying 100,000s of peptides, highly-multiplexed serology can provide epitope-resolved data across a wide variety of antigens, and therefore, this approach may be useful for clinical diagnostics, either directly or for the discovery of antigens that can then be adapted for use with less highly-multiplexed approaches. However, in this study, we did not attempt to calculate measures of sensitivity or specificity for the detection of specific virus exposures/immunizations. This is because our sample size for each specific immunogen was very small and because our knowledge of the infection histories of the characterized samples was incomplete.”

Reviewer #2 (Public repository details (Required)):

NGS as read-out but did not see accession numbers for SRA or BioProject for NCBI depositing. Needs this.

Our analysis workflow does not generate sample-specific fastq files, which would be needed for deposition into NCBI's Sequence Read Archive (SRA). However, we have added our raw, peptide-level counts for all samples as supplemental material. These raw counts are generated directly from the read level data (which is input as a single aggregate fastq file that contains data from multiple samples and often multiple assays and projects), and can be used to recreate all of the analyses presented in the paper.

Due to the large size of some of these files, they have been added to our OSF archive for this paper (<https://osf.io/8hygc/>, “Data” directory). A link to this repository is already present in the “Data availability statement”:

“All relevant data are within the paper, within Supporting Information files and/or have been deposited in OSF (<https://osf.io/8hygc/>).”

Reviewer #2 (Comments for the Author):

This manuscript describes use of protein A and protein G to perform highly multiplexed serology in 25 non-human mammals along with candidate validation and discovery work using the

multiplexed serology. The capacity of protein A and protein G to work in non-human mammals has been recognized for a long time, though has mostly been limited to model species. This paper tests a broad range of mammals that are relevant for zoonotic viral discovery and serological screening efforts as the authors note. It is also helpful to have a systematic study of rough affinities and best reagents in the context of highly multiplexed serology rather than, say, a monoclonal pulldown. A weakness of the study is the lack of formal determination of Kd's for binding of Fc and limitation of the study to protein A and protein G as capture reagents. The study also is a bit of an odd mishmash of data from the FLISA by mammal and control infection data to the discovery pulldown data in Figure 5, which is intriguing but perhaps not a complete story without the confirmation of the agent, though this could be considered beyond the scope of this paper, which is this reviewer's opinion.

We thank the reviewer for their enthusiasm for our manuscript, and we agree that the additional topics highlighted by the reviewer will be a great focus for future studies, but are beyond the scope of this manuscript.

-Authors should specifically note why linear regression used throughout the paper instead of 4PL regression for FLISA dilution data.

As suggested, we have added the following statement to our Methods section:

“The slope we measured should be roughly equivalent to the hill coefficient of a four parameter logistic curve. However, we found that the linear fit was more broadly applicable across the range of affinities that we were measuring, while using a consistent dilution series across samples (Fig. 1B).”

It would be nice to see plots such as Figure 5D in supplementary figures for any highly multiplexed pulldown that is performed throughout the paper (this includes Figure 4 pulldowns) to inform pure discovery efforts in any pulldown that is performed. The genome plots are fantastic, but it can somewhat hide the noise of the data if not all the peptides are plotted somewhere.

We agree. We have generated interactive scatter plots similar to Figure 5D for all of the assays presented in this study. These figures highlight the peptides that have been considered enriched for each sample and will allow readers to obtain additional information for these peptides (e.g., the virus species from which they were designed and the actual peptide sequences). Here is an example of one of these plots, along with the pop up window that appears whenever the user hovers over one of the enriched peptides:

We have added these supplemental figures to our OSF archive for this paper (<https://osf.io/8hygc/>, “Supporting Info/Individual Scatterplots”). These have been deposited as QIIME 2 visualization files (QZV format), which can be viewed using the qiime2view web interface (<https://view.qiime2.org/>). This is described within the “readme.txt” file present within the OSF archive.

We have also added the following to the results (new text is shown in **bold**):

“In total, we conducted at least one PepSeq assay for all of the non-human mammal species tested with our FLISA assay (n=25), and we detected ≥ 1 enriched peptide for at least one sample/library combination for every species (**interactive scatter plots highlighting the enriched peptides for each sample have been deposited in OSF: <https://osf.io/8hygc/>**.)”

I did not see any accession numbers or a BioProject for sequencing reads.

Please see above for description of data archival approach.

No minor edits, the paper is well-written.

Thank you.

September 6, 2022

Dr. Jason T Ladner
Northern Arizona University
1395 Knoles Drive
Coconino County
Flagstaff, AZ 86001

Re: Spectrum02873-22R1 (Highly-multiplexed serology for non-human mammals)

Dear Dr. Jason T Ladner:

Your manuscript has been accepted, and I am forwarding it to the ASM Journals Department for publication. You will be notified when your proofs are ready to be viewed.

Sincerely,

Ralph Tripp
Editor, Microbiology Spectrum

Journals Department
Supplemental Material: Accept
S1 Table: Accept